# Generating Lattice Non-invertible Symmetries

Weiguang Cao[a,b,1], Linhao Li[c,2], and Masahito Yamazaki[a,d,3]

[a]  Kavli Institute for the Physics and Mathematics of the Universe,
     University of Tokyo, Kashiwa, Chiba 277-8583, Japan
[b]  Department of Physics, Graduate School of Science,
     University of Tokyo, Tokyo 113-0033, Japan
[c]  Department of Physics and Astronomy, Ghent University,
     Krijgslaan 281, S9, B-9000 Ghent, Belgium
[d]  Trans-Scale Quantum Science Institute,
     University of Tokyo, Tokyo 113-0033, Japan

Lattice non-invertible symmetries have rich fusion structures and play important roles in understanding various exotic topological phases. In this paper, we explore methods to generate new lattice non-invertible transformations/symmetries from a given non-invertible seed transformation/symmetry. The new lattice non-invertible symmetry is constructed by composing the seed transformations on different sites or sandwiching a unitary transformation between the transformations on the same sites. In addition to known non-invertible symmetries with fusion algebras of Tambara-Yamagami $\mathbb{Z}_N \times \mathbb{Z}_N$ type, we obtain a new non-invertible symmetry in models with $\mathbb{Z}_N$ dipole symmetries. We name the latter the dipole Kramers-Wannier symmetry because it arises from gauging the dipole symmetry. We further study the dipole Kramers-Wannier symmetry in depth, including its topological defect, its anomaly and its associated generalized Kennedy-Tasaki transformation.

[1]weiguang.cao@ipmu.jp
[2]linhaoli601@163.com
[3]masahito.yamazaki@ipmu.jp

# 1 Introduction

**Lattice non-invertible symmetry:** Recently, the study of non-invertible symmetries in lattice models with tensor product Hilbert spaces has attracted much attention [1–6]. These works study the generalized symmetries in field theory initiated by the seminal paper [7]. While an ordinary symmetry is defined by topological operators with group-like fusion rules, there exist many examples of topological operators with more general fusion rules wherein an operator does not necessarily have an inverse. The so-called *non-invertible symmetry* has been extensively studied in field

theories [8–12] and lattice models [1,2,13–16] in $(1+1)d$, and later in higher dimensions [17–23] with potential applications in the particle phenomenology [24–26].[4]

One of the common methods to generate a non-invertible symmetry is to gauge a discrete Abelian group symmetry $G$ [8,27]. In $(1+1)d$, gauging a 0-form discrete symmetry $G$ leads to a gauged theory with a new 0-form symmetry $\hat{G} := \mathrm{Rep}(G)$, where $\mathrm{Rep}(G)$ is the representation category of the original group $G$. For a discrete Abelian group $G$, $\mathrm{Rep}(G)$ is again a discrete Abelian group and is isomorphic to $G$. A well-known example is the gauging of a 0-form $\mathbb{Z}_N$ symmetry. Suppose the $\mathbb{Z}_N$ symmetry is generated by an operator $\eta$ with $\eta^N = 1$. Let us denote the Kramers-Wannier (KW) duality transformation from the gauging of this $\mathbb{Z}_N$ symmetry as $\mathcal{D}$. The invertible and non-invertible operators follow the fusion algebras of the Tambara-Yamagami type $\mathrm{TY}(\mathbb{Z}_N)$ [28]:[5]

$$\eta \times \mathcal{D} = \mathcal{D} \times \eta = \mathcal{D}, \quad \mathcal{D} \times \mathcal{D} = \sum_{k=1}^{N} \eta^k. \tag{1.1}$$

For $N = 2$, this gives the fusion rules of the Ising CFT [29,30]. If the theory is further invariant under the gauging of the discrete Abelian group symmetry $G$, the non-invertible duality transformation will become a non-invertible symmetry.

For a lattice model with a discrete Abelian group symmetry $G$, we can also find a non-invertible symmetry by searching for models invariant under gauging the $G$ symmetry. However, non-invertible symmetries in lattice models have a richer structure than those in field theories. Even the lattice KW symmetry D from gauging a $\mathbb{Z}_2$ symmetry exhibits a fusion rule mixed with the one-site lattice translation operator T:

$$U \times \mathsf{D} = \mathsf{D} \times U = \mathsf{D}, \quad \mathsf{D} \times \mathsf{D} = (1+U)\mathsf{T}^{-1}, \quad U^2 = 1, \tag{1.2}$$

where $U$ generates the $\mathbb{Z}_2$ symmetry. This lattice KW symmetry (1.2) is therefore beyond the fusion category description [5]. This observation has been generalized to the subsystem non-invertible symmetry from gauging subsystem $\mathbb{Z}_2$ [31] and $\mathbb{Z}_N$ symmetry [32] in $(2+1)d$. It is desirable to have more examples of lattice non-invertible symmetries and further explore the rich structures in lattice non-invertible fusion algebras. Moreover, lattice non-invertible symmetries play an important role in understanding various exotic topological phases through generalized 't Hooft anomalies [13,33,34] and a generalized Landau paradigm [16,35–38].

In this paper, we study lattice non-invertible symmetries by composing a given non-invertible symmetry, which we call the *seed transformation*, with another invertible or non-invertible symmetry. This method has been very powerful in field theories. For example, in the $c = 1$ compact boson, there exist non-invertible symmetries composed of a $\mathbb{Z}_N$-symmetry gauging and the T-duality transformation. Another example is the non-invertible symmetry from the Adler-Bell-Jackiw (ABJ) anomaly [24,25], where the non-invertible rational-angle chiral rotation symmetry

---

[4]We apologize in advance for missing references and welcome suggestions for additional relevant references.

[5]In this paper, we will only focus on the fusion algebra and leave the discussion of other fusion data in future exploration. Therefore we omit the specification of the bicharacter and Frobenius-Schur indicator in the description of fusion category throughout this paper.

is constructed by stacking a $(2 + 1)d$ fractional quantum Hall state to a gauge-dependent conserved current. This method has also been studied in lattice models. The non-invertible symmetry of the cluster state is constructed by composing two KW duality operators that act separately on even and odd sites [6, 39]. Furthermore, a subsystem non-invertible symmetry is constructed by the product of KW transformations on every line, a global Hadamard gate and the product of KW transformations on every column in [32]. In this paper, we continue to explore this construction in models with dipole symmetries.

**Dipole symmetry:** Besides non-invertible symmetry, lattice models are playgrounds for other exotic symmetries like subsystem symmetry [40–54],[6] fractal symmetry [60] and multipole symmetry [61–71]. Models with these exotic symmetries attract much interest because they admit quasi-particles called fractons with restricted mobilities [72–82] and have UV-IR mixing in the continuum limit [47,83–88]. In recent years, dipole symmetry has been studied in new topological insulators [89–91], exotic quantum liquids [92–95], systems with non-ergodicity [96–99], hydrodynamics with dipole conservation [100–103] and systems with anomalous diffusion [104–106].

In field theories with dipole symmetries, the charge $Q$ and the dipole $D_i$ in the $x_i$ direction are conserved quantities:

$$Q := \int_{\text{space}} J_0, \quad D_i := \int_{\text{space}} x_i J_0, \tag{1.3}$$

where $J_0$ is the time component of the conserved current. The dipole conservation constrains the mobility of a single charge. If we further impose translational invariance generated by conserved momentum $P_i$, the dipole symmetry mixes with the translation symmetry

$$[P_j, D_k] = -i\delta_{j,k}Q. \tag{1.4}$$

One can consider the simplest example with $U(1)$ dipole shift symmetry in $(1 + 1)d$ [64]

$$S = \oint d\tau dx \left( \frac{\mu_0}{2}(\partial_\tau \phi)^2 + \frac{1}{2\mu}(\partial_x^2 \phi)^2 \right), \quad \phi \sim \phi + 2\pi, \tag{1.5}$$

which is invariant under the constant and dipole shifts:

$$\phi(\tau, x) \rightarrow \phi(\tau, x) + c + c_x x, \quad c, c_x \in U(1). \tag{1.6}$$

To gauge the symmetry, we need to couple the system with tensor gauge fields $(A_\tau, A_{xx})$ with the exotic gauge transformation

$$A_\tau \rightarrow A_\tau + \partial_\tau \alpha(\tau, x), \quad A_{xx} \rightarrow A_{xx} + \partial_x^2 \alpha(\tau, x), \tag{1.7}$$

to cancel the contribution from shifting $\phi(\tau, x)$ by an arbitrary function $\alpha(\tau, x) \in U(1)$. In the literature, the corresponding gauge theory is called the $U(1)$ dipole gauge symmetry although both charge and dipole symmetry have been gauged.

---

[6]Subsystem symmetry is also known as gauge-like symmetry in previous literature [55–59]

The above analysis works in parallel for a lattice model with a $\mathbb{Z}_N$ dipole symmetry. Here, the charge and the dipole operators are

$$\eta_Q := \prod_{i=1}^{L} X_i, \quad \eta_D := \prod_{i=1}^{L} (X_i)^i, \quad \eta_Q^N = \eta_D^N = 1, \quad \mathsf{T}\eta_D = \eta_Q \eta_D \mathsf{T}, \tag{1.8}$$

where $X_i, Z_i$ are $\mathbb{Z}_N$ Pauli operators acting on site $i$. The discussion of boundary conditions is subtle for a dipole symmetry, and we will for simplicity assume a periodic boundary condition with lattice size $L \equiv 0 \bmod N$ throughout this paper. The dipole symmetry acts as

$$\eta_D : \quad Z_i \to \omega^i Z_i, \quad \omega := \exp(2\pi i/N), \tag{1.9}$$

and the simplest symmetric interaction is $Z_{i-1}(Z_i^\dagger)^2 Z_{i+1}$.

In this paper, we illustrate the procedure of simultaneously gauging the charge and dipole symmetries on the lattice. We find that this gauging implements a non-invertible duality transformation, which we call the *dipole KW transformation*. Similar to the case of field theory, we will interchangeably use gauging the dipole symmetry and gauging the $\mathbb{Z}_N^Q \times \mathbb{Z}_N^D$ symmetry.

**Non-invertible dipole KW symmetry:** Although there are extensive studies of non-invertible symmetry and exotic symmetry, the two types of symmetries are often treated separately, and non-invertible symmetries in models with exotic symmetry are still rarely explored. There are a few examples in lattice models with subsystem symmetry [31, 32] and field theories with exotic symmetry [107]. Studying non-invertible symmetries in models with exotic symmetry will deepen our understanding of non-invertible symmetries and their fusion structures through more non-trivial and exotic examples. For instance, the fusion of subsystem non-invertible symmetry forms a grid operator which is a sum of non-topological invertible operators. In this paper, we further construct and study the lattice non-invertible symmetry in $(1+1)d$ spin models with dipole symmetry.

By conjugating the global Hadamard gate with the ordinary KW transformation, we obtain a new non-invertible transformation, the *dipole KW transformation*, that exchanges the transverse field and dipole interaction. Surprisingly, the fusion algebra of the dipole KW transformation mixes with the charge conjugation symmetry instead of the lattice translation,[7] further extending our understanding of the lattice non-invertible structure. In Table. 1, we list different fusion structures that are studied in this paper.

The dipole KW transformation can be obtained from gauging the dipole symmetry. We then study the topological defects by a half-gauging on the spin chain and investigate the anomaly of the dipole KW symmetry. In particular, the dipole Ising model with $N = 3$ at two self-dual points admits an anomalous non-invertible symmetry. By gauging the ordinary $\mathbb{Z}_3$ symmetry, the model is mapped to the $\mathbb{Z}_3$ anisotropic XZ model (or quantum torus chain [108]). The anomaly prohibits the gapped phase with a unique ground state, which is consistent with the gapless phase and the first order phase transition from numerical calculation [109, 110]. Based on the non-invertible

---

[7]In the appendix of [32], the charge conjugation also appears in the fusion rule of the subsystem non-invertible symmetry from gauging $\mathbb{Z}_N$ subsystem symmetry. But in that case, the fusion rule also mixes with lattice translation.

| Duality transformation | self-fusion | Pattern |
| --- | --- | --- |
| gauging $\mathbb{Z}_N : \mathsf{D}$ | $(\sum_{k=1}^{N} \eta^k)\mathsf{T}^{-1}$ | Mixed with lattice translation $\mathsf{T}$ |
| gauging $\mathbb{Z}_N \times \mathbb{Z}_N : \mathsf{D}_o^\dagger \mathsf{D}_e$ | $(\sum_{k=1}^{N} \eta_o^k)(\sum_{k=1}^{N} \eta_e^k)$ | No mixing |
| gauging $\mathbb{Z}_N^Q \times \mathbb{Z}_N^D : \mathsf{D}^\dagger U^H \mathsf{D}$ | $(\sum_{k=1}^{N} \eta_Q^k)(\sum_{k=1}^{N} \eta_D^k)\mathsf{C}$ | Mixed with charge conjugation $\mathsf{C}$ |

Table 1: Different Fusion structures in $(1+1)d$ models in this paper.

symmetry, we define the Kennedy-Tasaki (KT) transformations [111–119] associated with dipole symmetries, as a duality relating a dipole spontaneously symmetry breaking (SSB) phase and a dipole symmetry protected topological (SPT) phase. Following the construction in [3], KT transformation is a composition of the KW duality transformation and the invertible transformation that stacks an SPT to the system. Recently, KT transformations have been applied to studying and classifying non-trivial gapped and gapless phases [6, 32, 120–125].

**The structure of this paper:** In Sec. 2 we first review the construction of a non-invertible transformation with $\mathrm{TY}(\mathbb{Z}_N \times \mathbb{Z}_N)$ type fusion algebra and define the dipole KW transformation from the ordinary KW transformation as the seed transformation. In Sec. 3, we study the dipole KW symmetry operator and defect by gauging the dipole symmetry. We also study the mapping of symmetry-twist sectors of the charge and dipole symmetry during the dipole KW transformation. We study the anomaly condition for the dipole KW symmetry in Sec. 4 and the generalized Kennedy-Tasaki transformation in Sec. 5. Finally, we conclude and point out future directions in Sec. 6.

**Note added:** After the completion of this manuscript, we became aware of an independent related work [126] on gauging finite modulated symmetries on spin chains, which will appear on arXiv soon.

## 2 Generating lattice non-invertible symmetries

In this section, we show how to generate more lattice non-invertible symmetries in $(1+1)d$ lattice models from a given seed lattice non-invertible transformation. We study examples where the seed transformation is the KW transformation from gauging a $\mathbb{Z}_N$ symmetry. By composing the seed transformation acting on different sites or with other invertible transformations, we either generate a known symmetry with fusion algebras of Tambara-Yamagami type $\mathrm{TY}(\mathbb{Z}_N \times \mathbb{Z}_N)$, or a totally new non-invertible symmetry which appears in models with dipole symmetries.

We work on a spin chain of $L$ sites with the periodic boundary condition. On each site there is a qudit $|s_i\rangle_i \in \mathcal{H}_i, s_i \in \mathbb{Z}_N$. The total Hilbert space $\mathcal{H}$ is a tensor product of the local Hilbert

space $\mathcal{H}_i$ on each site $i$. We use $|\{s_i\}\rangle := \otimes_{i=1}^{L} |s_i\rangle_i \in \mathcal{H}$ for a state with qudit $s_i$ at site $i$. The Pauli operators acting on each site are defined as

$$Z_i := \sum_{s_i=1}^{N} \omega^{s_i} |s_i\rangle_i \langle s_i|, \quad X_i := \sum_{s_i=1}^{N} |s_i+1\rangle_i \langle s_i|, \quad \omega := \exp(2\pi i/N). \tag{2.1}$$

The Pauli operators are examples of local unitary operators and satisfy the following algebra

$$Z_i^N = X_i^N = 1, \quad Z_i X_i = \omega X_i Z_i, \quad [Z_i, X_j] = 0 \quad \text{for } i \neq j. \tag{2.2}$$

## 2.1 The seed transformation

In this paper, we use the KW transformation D of gauging the $\mathbb{Z}_N$ symmetry as a seed transformation to generate more lattice non-invertible symmetries. The $\mathbb{Z}_N$ symmetry is generated by

$$\eta := \prod_{i=1}^{L} X_i, \quad \eta^N = 1. \tag{2.3}$$

A Hamiltonian with $\mathbb{Z}_N$ symmetry can be constructed from $\mathbb{Z}_N$-singlet operators $\{Z_i^\dagger Z_{i+1}, X_i\}$. A typical example is the $\mathbb{Z}_N$ clock model with a transverse field

$$H_{\text{clock}} = -g^{-1} \sum_{i=1}^{L} Z_i^\dagger Z_{i+1} - g \sum_{i=1}^{L} X_i + (\text{h.c.}), \tag{2.4}$$

where h.c. means the Hermitian conjugate. This model contains the transverse Ising model ($N = 2$) and three states Potts model ($N = 3$) as special cases. In this paper, we only focus on Hermitian models which automatically have the charge conjugation symmetry $\mathbb{Z}_N^C$ generated by[8]

$$\mathsf{C} = \sum_{\{s_j\}} |\{-s_j\}\rangle \langle \{s_j\}| : \quad Z_i \to Z_i^\dagger, \quad X_i \to X_i^\dagger, \quad i = 1, ..., L, \tag{2.6}$$

where we sum over all possible spin configurations of $\{s_i\}$.

The KW transformation D acts on the operators as

$$\mathsf{D}: \quad X_i \to Z_{i-1}^\dagger Z_i, \quad Z_{i-1}^\dagger Z_i \to X_{i-1}. \tag{2.7}$$

which preserves the locality of a $\mathbb{Z}_N$ invariant Hamiltonian. The KW transformation maps the clock model (2.4) from a symmetric gapped phase with a unique ground state ($g \gg 1$) to a symmetry breaking phase with $N$ degenerate vacua ($0 \leq g \ll 1$) and vice versa. At the critical point ($g = 1$), the KW transformation becomes a non-invertible symmetry.

---

[8]Throughout this paper, for a general (unitary or non-unitary) operator $\mathcal{O}$, its action on other operators $O_a$ is

$$\mathcal{O}O_a = O_a'\mathcal{O}: \quad O_a \to O_a'. \tag{2.5}$$

The KW transformation has several realizations in the literature [5]. In Appendix A we illustrate how to get the KW transformation D from gauging the $\mathbb{Z}_N$ symmetry and study the topological defect of the KW duality symmetry. Such the KW transformation can be rewritten in the bilinear phase map (BPM) representation [127]

$$\mathsf{D} = \sum_{\{s_i\},\{s'_i\}} \omega^{\sum_{i=1}^L (s_{i-1}-s_i)s'_{i-1}} |\{s'_i\}\rangle \langle\{s_i\}|, \tag{2.8}$$

where we sum over all possible spin configurations of $\{s_i\}$ and $\{s'_i\}$. In the BPM representation, it is convenient to compute various operator relations. The action on the Pauli operators is

$$\mathsf{D}\left(Z_{j-1}^\dagger Z_j\right) = X_{j-1}\mathsf{D}, \quad \mathsf{D}X_j = \left(Z_{j-1}^\dagger Z_j\right)\mathsf{D}. \tag{2.9}$$

The KW operator D commutes with the translation operator T which acts on states by a shift of the value of qudit to one site right

$$\mathsf{T}|\{s_i\}\rangle := \bigotimes_{i=1}^L |s_{i-1}\rangle_i, \tag{2.10}$$

and their fusion gives the Hermitian conjugate $\mathsf{D}^\dagger$

$$\mathsf{D}^\dagger = \mathsf{D}\mathsf{T} = \mathsf{T}\mathsf{D}. \tag{2.11}$$

The fusion algebra of the invertible operator $\eta$, C, the non-invertible operator D and the translation operator T is

$$\mathsf{C}\mathsf{D} = \mathsf{D}\mathsf{C}, \quad \eta\mathsf{D} = \mathsf{D}\eta = \mathsf{D}, \quad \mathsf{D} \times \mathsf{D} = \left(\sum_{k=1}^N \eta^k\right)\mathsf{T}^{-1}, \quad \mathsf{D}^\dagger \times \mathsf{D} = \sum_{k=1}^N \eta^k,$$
$$\eta\mathsf{C} = \mathsf{C}\eta^\dagger, \quad \mathsf{T}\eta = \eta\mathsf{T}, \quad \mathsf{T}\mathsf{C} = \mathsf{C}\mathsf{T}, \quad \mathsf{C}^2 = 1, \quad \mathsf{T}^L = 1. \tag{2.12}$$

The detailed computation of (2.9), (2.11) and non-invertible fusion rules in (2.12) can be found in Appendix C. Other fusion rules follow simply from the definition. Because the invertible operators are defined in a translational invariant way, they commute with the translation operator T. D absorbs $\eta$ because of (2.9). The charge conjugation C commutes with the non-invertible KW operator D because

$$\mathsf{C}\mathsf{D} = \sum_{\{s_i\},\{s'_i\}} \omega^{\sum_{i=1}^L (s_{i-1}-s_i)(-s'_{i-1})} |\{s'_i\}\rangle \langle\{s_i\}| = \sum_{\{s_i\},\{s'_i\}} \omega^{\sum_{i=1}^L (-(s_{i-1}-s_i))s'_{i-1}} |\{s'_i\}\rangle \langle\{s_i\}| = \mathsf{D}\mathsf{C}. \tag{2.13}$$

Because of the appearance of lattice translation transformation, the subalgebra $\{\eta, \mathsf{D}\}$ differs from the fusion algebra $\mathrm{TY}(\mathbb{Z}_N)$ (1.1).

## 2.2 TY($\mathbb{Z}_N \times \mathbb{Z}_N$) type fusion algebra

In this subsection, we show how to generate lattice non-invertible transformations by composing the seed transformations acting on different sites (or different degrees of freedom in a unit cell). We work out the example of composing the KW transformations on even and odd sites.

In this particular example, we assume that the number of sites is even ($L \equiv 0 \mod 2$) and the model has a $\mathbb{Z}_N^o \times \mathbb{Z}_N^e$ symmetry where $\mathbb{Z}_N^o$ acting on the odd sites and $\mathbb{Z}_N^e$ acting on the even sites. This symmetry is generated by

$$\eta_o = \prod_{i=1}^{L/2} X_{2i-1}, \quad \eta_e = \prod_{i=1}^{L/2} X_{2i}. \tag{2.14}$$

The model with $\mathbb{Z}_N^o \times \mathbb{Z}_N^e$ symmetry is constructed from the symmetry-singlet operators $\{Z_{i-1}^\dagger Z_{i+1}, X_i\}$ with examples of Ising type model

$$H_{\text{Ising-type}} = -g^{-1} \sum_{i=1}^{L} Z_{i-1}^\dagger Z_{i+1} - g \sum_{i=1}^{L} X_i + (\text{h.c.}), \tag{2.15}$$

and the $\mathbb{Z}_N$ cluster model

$$H_{\text{cluster}} = -\sum_{i=1}^{L} Z_{i-1}^\dagger X_i Z_{i+1} + (\text{h.c.}). \tag{2.16}$$

The seed transformations are the KW transformations acting on even and odd sites

$$\begin{aligned}
\mathsf{D}_o : \quad & X_{2i+1} \to Z_{2i-1}^\dagger Z_{2i+1}, \quad Z_{2i-1}^\dagger Z_{2i+1} \to X_{2i-1}, \\
\mathsf{D}_e : \quad & X_{2i} \to Z_{2i-2}^\dagger Z_{2i}, \quad Z_{2i-2}^\dagger Z_{2i} \to X_{2i-2}.
\end{aligned} \tag{2.17}$$

Since they acts on different sites, $\mathsf{D}_o, \mathsf{D}_e$ commute. We can find two more non-invertible transformations, by composing $\mathsf{D}_o, \mathsf{D}_e$ and the lattice translation $\mathsf{T}$

$$\begin{aligned}
\tilde{\mathsf{D}} = \mathsf{T}\mathsf{D}_o\mathsf{D}_e : \quad & X_i \to Z_{i-1}^\dagger Z_{i+1}, \quad Z_{i-1}^\dagger Z_{i+1} \to X_i, \\
\tilde{\mathsf{D}}' = \mathsf{D}_o\mathsf{D}_e : \quad & X_i \to Z_{i-2}^\dagger Z_i, \quad Z_{i-2}^\dagger Z_i \to X_{i-2},
\end{aligned} \tag{2.18}$$

with fusion algebras

$$\begin{aligned}
\tilde{\mathsf{D}} \times \tilde{\mathsf{D}} &= \left(\sum_{k=1}^{N} \eta_o^k\right)\left(\sum_{k=1}^{N} \eta_e^k\right), \\
\tilde{\mathsf{D}}' \times \tilde{\mathsf{D}}' &= \left(\sum_{k=1}^{N} \eta_o^k\right)\left(\sum_{k=1}^{N} \eta_e^k\right) \mathsf{T}^{-2}.
\end{aligned} \tag{2.19}$$

Here are some comments about the new non-invertible transformations

1. For $N = 2$ both $\tilde{\mathsf{D}}$ and $\tilde{\mathsf{D}}'$ are non-invertible symmetries for the $\mathbb{Z}_2$ cluster state [6]. The $\tilde{\mathsf{D}}$ symmetry belongs to the $\mathrm{Rep}(D_8)$ fusion category symmetry. The $\tilde{\mathsf{D}}'$ symmetry becomes the $\mathrm{Rep}(H_8)$ fusion category symmetry in the continuum limit.[9]

2. Because the fusion of $\tilde{\mathsf{D}}$ does not mix with lattice translation, the subalgebra of $\eta_o, \eta_e, \tilde{\mathsf{D}}$ forms the fusion algebra of $\mathrm{TY}(\mathbb{Z}_N \times \mathbb{Z}_N)$ in the lattice level.

3. For general $N$, there might not exist an SPT phase that is invariant under $\tilde{\mathsf{D}}$ or $\tilde{\mathsf{D}}'$ symmetry. This fact can be used as a diagnosis of the anomaly of the non-invertible symmetries.

4. It is straightforward to generalize this method and find new non-invertible transformations and symmetries in lattice models with Abelian group $G_1 \times G_2$ symmetry. The seed transformations are gauging the $G_1$ symmetry and gauging the $G_2$ symmetry.

## 2.3  Dipole Kramers-Wannier transformation

In this subsection, we explore yet another method to generate the lattice non-invertible transformation through sandwiching unitary operators by seed transformations acting on the same sites. This idea comes from an observation [128] that gauging the $\mathbb{Z}_N$ symmetry $\eta = \prod_{i=1}^{L} X_i$ of the XZ model

$$H_{\mathrm{XZ}} = -g^{-1} \sum_{i=1}^{L} X_i^\dagger X_{i+1} - g \sum_{i=1}^{L} Z_i^\dagger Z_{i+1} + (\text{h.c.}) \tag{2.20}$$

gives the dipole Ising model with $\mathbb{Z}_N^D$ dipole symmetry

$$H_{\text{dipole-Ising}} = -g^{-1} \sum_{i=1}^{L} Z_{i-1}(Z_i^\dagger)^2 Z_{i+1} - g \sum_{i=1}^{L} X_i + (\text{h.c.}). \tag{2.21}$$

Actually the model has a larger symmetry $\left(\mathbb{Z}_N^Q \times \mathbb{Z}_N^D\right) \rtimes \mathbb{Z}_2^C$ symmetry where the superscripts $Q, D, C$ means charge, dipole and charge conjugation. The generators of the charge and dipole symmetry are

$$\eta_Q := \prod_{i=1}^{L} X_i, \quad \eta_D := \prod_{i=1}^{L} (X_i)^i. \tag{2.22}$$

For simplicity, we assume $L \equiv 0 \bmod N$ to have a full $\mathbb{Z}_N^D$ symmetry.

Starting with the seed transformation D by gauging the charge symmetry $\mathbb{Z}_N^Q$, we define the *dipole KW transformation*

$$\hat{\mathsf{D}} := \mathsf{T}\mathsf{D}U^H\mathsf{D} = \mathsf{D}^\dagger U^H\mathsf{D}, \tag{2.23}$$

---

[9]$\mathrm{Rep}(D_8)$ and $\mathrm{Rep}(H_8)$ have the same fusion algebra but different bicharacters. The difference in lattice transformations is a reflection of the difference of the bicharacter between the two. In this paper we only focus on the fusion algebra and leave the study of other fusion data (such as bicharacter) in future work.

where

$$U^H := \prod_{i=1}^{L} U_i^H, \quad U_i^H = \frac{1}{\sqrt{N}} \sum_{\alpha,\beta=1}^{N} \omega^{-\alpha\beta} |\beta\rangle \langle\alpha| : \quad X_i \to Z_i^\dagger, \quad Z_i \to X_i, \tag{2.24}$$

is the product of $\mathbb{Z}_N$ Hadamard gate $U_i^H$ on every site and can be viewed as the "half-charge conjugation" because $\mathsf{C} = (U^H)^2$. From the decomposition (2.23), the dipole KW transformation first maps the dipole Ising model to the XZ model, then exchanges $X_i^\dagger X_{i+1}$ and $Z_i^\dagger Z_{i+1}$, and finally maps the model back to the dipole Ising model. The net action is

$$\hat{\mathsf{D}} : \quad Z_{i-1}(Z_i^\dagger)^2 Z_{i+1} \to X_i, \quad X_i \to (Z_{i-1}(Z_i^\dagger)^2 Z_{i+1})^\dagger, \tag{2.25}$$

which interchanges the dipole-interaction terms and the transverse-field terms. The above transformations are summarized in the duality web in Fig. 1.

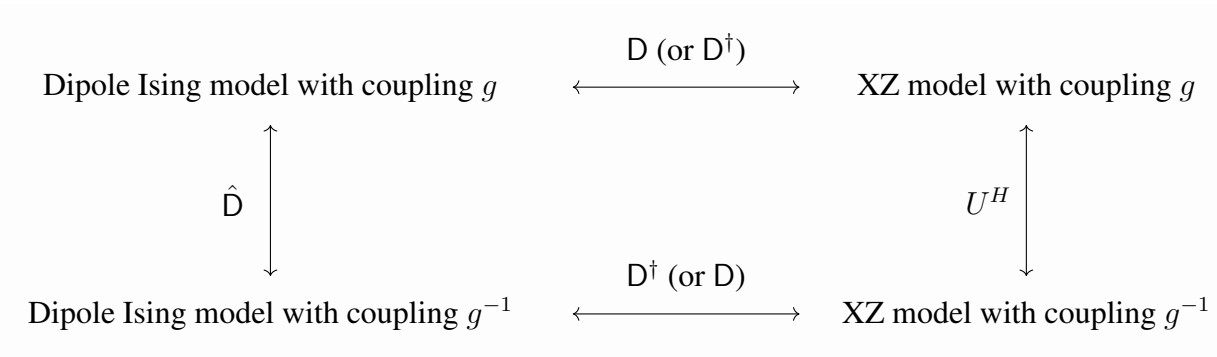

Figure 1: Duality web of the dipole Ising model and the XZ model.

In the next section, we will interpret the dipole KW transformation as gauging of both the charge and dipole symmetry and study the transformation in detail. The fusion algebra of $\eta_Q, \eta_D, \hat{\mathsf{D}}$ can already be computed from the decomposition of $\hat{\mathsf{D}}$ (2.23) and the property of the seed transformation $\mathsf{D}$. For example, the non-invertible fusion rule is

$$\hat{\mathsf{D}} \times \hat{\mathsf{D}} = (\mathsf{D}^\dagger U^H \mathsf{D})(\mathsf{D}^\dagger U^H \mathsf{D}) = \mathsf{D}^\dagger U^H \left( \sum_{k=1}^{N} \eta_Q^k \right) U^H \mathsf{D}$$

$$= \mathsf{D}^\dagger \left( \sum_{k=1}^{N} \left( \prod_{i=1}^{L} Z_i \right)^k \mathsf{C} \right) \mathsf{D} = \mathsf{D}^\dagger \mathsf{D} \left( \sum_{k=1}^{N} \eta_D^k \right) \mathsf{C} = \left( \sum_{k=1}^{N} \eta_Q^k \right) \left( \sum_{k=1}^{N} \eta_D^k \right) \mathsf{C}, \tag{2.26}$$

where we used

$$\prod_{i=1}^{L} Z_i = \prod_{i=1}^{L} \left( Z_{i-1} Z_i^\dagger \right)^i, \tag{2.27}$$

with the assumption $L \equiv 0 \bmod N$. The full fusion algebra of the invertible operators $\eta_Q, \eta_D, \mathsf{C}$, the non-invertible operator $\hat{\mathsf{D}}$ and the translation operator $T$ is

$$\eta_Q \hat{\mathsf{D}} = \eta_D \hat{\mathsf{D}} = \hat{\mathsf{D}} \eta_Q = \hat{\mathsf{D}} \eta_D = \hat{\mathsf{D}}, \quad \hat{\mathsf{D}} \times \hat{\mathsf{D}} = \left(\sum_{k=1}^{N} \eta_Q^k\right)\left(\sum_{k=1}^{N} \eta_D^k\right) \mathsf{C}, \quad \hat{\mathsf{D}}^\dagger = \mathsf{C}\hat{\mathsf{D}} = \hat{\mathsf{D}}\mathsf{C},$$

$$\mathsf{C}\eta_Q = \eta_Q^\dagger \mathsf{C}, \quad \mathsf{C}\eta_D = \eta_D^\dagger \mathsf{C}, \quad \mathsf{T}\eta_Q = \eta_Q \mathsf{T}, \quad \mathsf{T}\eta_D = \eta_Q^\dagger \eta_D \mathsf{T}, \quad \mathsf{C}^2 = 1, \quad \mathsf{T}^L = 1. \tag{2.28}$$

Here are a few comments about the dipole KW transformation:

1. In contrast with the ordinary KW transformation, implementing the dipole KW transformation $\hat{\mathsf{D}}$ twice does not involve a *lattice translation* $\mathsf{T}$ but a *charge conjugation operation* $\mathsf{C}$. The fusion subalgebra of $\eta_Q, \eta_Q, \hat{\mathsf{D}}, \mathsf{C}$ is new and is different from the fusion algebra of $\mathrm{TY}(\mathbb{Z}_N \times \mathbb{Z}_N)$. The $\mathbb{Z}_2$ charge conjugation appears in the non-invertible fusion rule and acts non-trivially on the invertible operators $\eta_Q, \eta_D$. For $N = 2$, the charge conjugation becomes the identity operator and the fusion subalgebra of $\eta_Q, \eta_Q, \hat{\mathsf{D}}$ becomes the fusion algebra of $\mathrm{Rep}(D_8)$.

2. For dipole Ising model at self dual point ($g = 1$), neither the ordinary KW transformation $\mathsf{D}$ nor the global Hadamard gate $U^H$ is a symmetry, while the dipole KW transformation (2.23) is a symmetry.

3. SPT phases protected by dipole symmetries in $(1+1)d$ have been classified in [129]. We will study the anomaly of the dipole KW symmetry by checking the invariance of dipole SPT phase under the dipole KW transformation for general $N$ in Section 4.

4. By repeating the sandwiching procedure (2.23), we can in general find non-invertible transformations in models with multipole symmetries. For example, the quadruple KW transformation is

$$\mathsf{D}U^H \mathsf{D}^\dagger U^H \mathsf{D}: \quad X_i \to Z_{i-2}(Z_{i-1}^\dagger)^3 Z_i^3 Z_{i+1}^\dagger, \quad Z_{i-2}(Z_{i-1}^\dagger)^3 Z_i^3 Z_{i+1}^\dagger \to X_{i-1}, \tag{2.29}$$

where the quadruple interaction is invariant under the quadruple symmetry operator

$$\eta_{\mathrm{Qu}} := \prod_i X_i^{i^2}. \tag{2.30}$$

# 3  Dipole Kramers-Wannier symmetry

In this section, we study the dipole KW symmetry in detail. We first show that the dipole KW transformation $\hat{\mathsf{D}}$

$$\hat{\mathsf{D}}: \quad Z_{i-1}(Z_i^\dagger)^2 Z_{i+1} \to X_i, \quad X_i \to (Z_{i-1}(Z_i^\dagger)^2 Z_{i+1})^\dagger, \tag{3.1}$$

comes from gauging both the charge and dipole symmetry. We then study the topological interface and defect of the dipole KW transformation by a half-gauging on the spin chain. Finally, using the BPM representation, we study how the symmetry and twist sectors of both the charge and the dipole symmetries map into each other under this non-invertible transformation.

## 3.1 Duality transformation: gauging on the whole spin chain

We gauge both the charge and dipole symmetry in the following steps. First we enlarge the Hilbert space by introducing gauge variables on the site. Then we impose the Gauss law constraints to project to the gauge invariant sectors. Finally we rename the variables to match with the original theory.

We illustrate the above steps in the transverse dipole Ising model

$$H_{\text{dipole-Ising}} = -g^{-1} \sum_{i=1}^{L} Z_{i-1}(Z_i^\dagger)^2 Z_{i+1} - g \sum_{i=1}^{L} X_i + \text{(h.c.)}. \tag{3.2}$$

Introducing another set of Pauli operators $\tilde{X}_i, \tilde{Z}_i$ on each site as gauge variables leads to the gauged Hamiltonian

$$H_{\text{gauged}} = -g^{-1} \sum_{i=1}^{L} \tilde{X}_i Z_{i-1}(Z_i^\dagger)^2 Z_{i+1} - g \sum_{i=1}^{L} X_i + \text{(h.c.)}. \tag{3.3}$$

The gauge variables $\tilde{X}_i, \tilde{Z}_i$ commutes with the original Pauli operators $X_i, Z_i$. The gauged theory is equivalent to coupling two isolated chains on top of each other, with an enlarged Hilbert space $\tilde{\mathcal{H}}$ with dimension $N^{2L}$. Besides the original global symmetry, the gauged theory also has a set of additional gauge symmetries generated by

$$G_i = X_i \tilde{Z}_{i-1}(\tilde{Z}_i^\dagger)^2 \tilde{Z}_{i+1}, \quad [G_i, H_{\text{gauged}}] = 0, \quad \forall i. \tag{3.4}$$

We require that the physical state $|\psi\rangle$ is gauge invariant

$$G_i |\psi\rangle = |\psi\rangle, \tag{3.5}$$

and we impose Gauss law constraints

$$G_i = X_i \tilde{Z}_{i-1}(\tilde{Z}_i^\dagger)^2 \tilde{Z}_{i+1} = 1, \quad \rightarrow \quad X_i = (\tilde{Z}_{i-1}(\tilde{Z}_i^\dagger)^2 \tilde{Z}_{i+1})^\dagger, \quad \forall i. \tag{3.6}$$

Changing to a new set of variables

$$\hat{Z}_i := \tilde{Z}_i, \quad \hat{X}_i := \tilde{X}_i Z_{i-1}(Z_i^\dagger)^2 Z_{i+1}, \tag{3.7}$$

the gauged Hamiltonian becomes

$$-g^{-1} \sum_{i=1}^{L} \hat{X}_i - g \sum_{i=1}^{L} (\hat{Z}_{i-1}(\hat{Z}_i^\dagger)^2 \hat{Z}_{i+1})^\dagger + \text{(h.c.)}. \tag{3.8}$$

Finally, by renaming the variables

$$\hat{Z}_i \to Z_i, \quad \hat{X}_i \to X_i, \tag{3.9}$$

we recover the original Hamiltonian with the change of coupling $g \to g^{-1}$. The whole process gives the dipole KW transformation $\hat{\mathsf{D}}$

$$\hat{\mathsf{D}}: \quad Z_{i-1}(Z_i^\dagger)^2 Z_{i+1} \to X_i, \quad X_i \to (Z_{i-1}(Z_i^\dagger)^2 Z_{i+1})^\dagger. \tag{3.10}$$

When the theory is self-dual with $g = 1$, $\hat{\mathsf{D}}$ becomes a non-invertible symmetry.

## 3.2 Duality defect: half-space gauging on the spin chain

In this subsection, we construct the duality defect corresponding to the dipole KW transformation by a half-space gauging on the spin chain. The half-space gauging creates a topological interface which further becomes a topological defect when the model is invariant under the gauging. We also show the local unitary gates that move and fuse the duality defect. The defect, its movement and fusion are shown in Fig. 2.

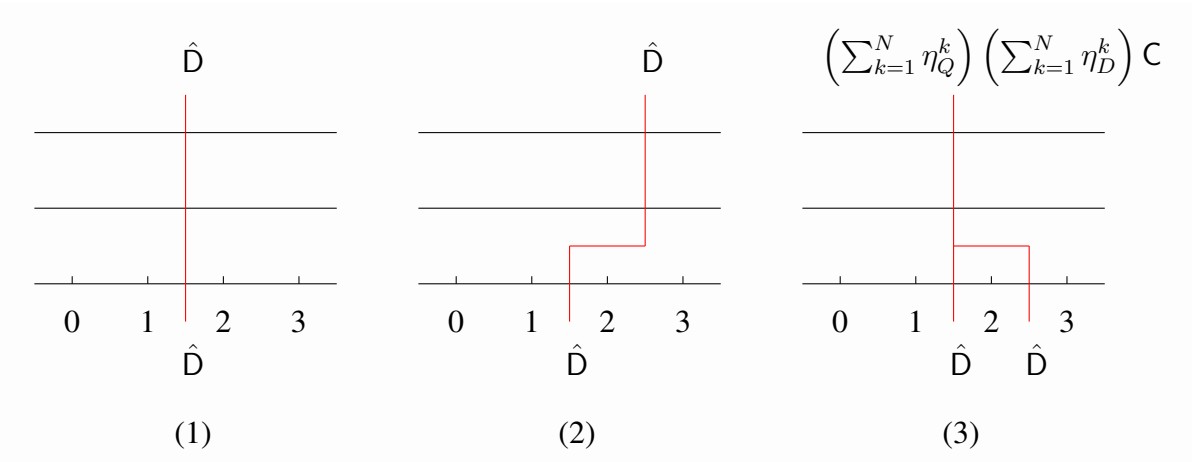

Figure 2: (1) Half-gauging creates a topological interface/defect $\hat{\mathsf{D}}$ located at link $(1, 2)$; (2) movement operator $U_{\hat{\mathsf{D}}}^2 = CZ_{2,3}(CZ_{2,l}^\dagger)^2(U_2^H)^\dagger S_{2,l}$ moves the interface/defect from $(1, 2)$ to $(2, 3)$; (3) Fusion operator $\lambda_{\hat{\mathsf{D}}\otimes\hat{\mathsf{D}}}^{(1,2,3)} = CZ_{3,l_2}^\dagger CZ_{2,l_1}^\dagger (CZ_{2,l_2})^2 S_{2,l_2}$ fuses the defects at $(1, 2)$ and $(2, 3)$.

**Half-space gauging**

Consider the transverse dipole Ising model on an infinite chain and gauge the dipole symmetry for sites $i \geq 1$. The half-gauged Hamiltonian is

$$H_{\text{half-gauged}} = -g^{-1}\sum_{i\leq 0} Z_{i-1}(Z_i^\dagger)^2 Z_{i+1} - g^{-1}\sum_{i\geq 1}\tilde{X}_i Z_{i-1}(Z_i^\dagger)^2 Z_{i+1} - g\sum_i X_i + (\text{h.c.}). \tag{3.11}$$

Imposing the Gauss law constraints that commute with the half-gauged Hamiltonian

$$G_i = X_i \tilde{Z}_{i-1} (\tilde{Z}_i^\dagger)^2 \tilde{Z}_{i+1} = 1, \quad i \geq 2, \tag{3.12}$$

then (3.11) becomes

$$
\begin{aligned}
H_{\text{half-gauged}} = & - g^{-1} \sum_{i \leq 0} Z_{i-1} (Z_i^\dagger)^2 Z_{i+1} - g \sum_{i \leq 0} X_i \\
& - g^{-1} \tilde{X}_1 Z_0 (Z_1^\dagger)^2 Z_2 - g X_1 - g^{-1} \tilde{X}_2 Z_1 (Z_2^\dagger)^2 Z_3 - g \tilde{Z}_1 (\tilde{Z}_2^\dagger)^2 \tilde{Z}_3 \\
& - g^{-1} \sum_{i \geq 3} \tilde{X}_i Z_{i-1} (Z_i^\dagger)^2 Z_{i+1} - g \sum_{i \geq 3} \tilde{Z}_{i-1} (\tilde{Z}_i^\dagger)^2 \tilde{Z}_{i+1} + (\text{h.c.}).
\end{aligned}
\tag{3.13}
$$

We can further introduce a new set of Pauli variables

$$
\begin{aligned}
Z_l &:= \tilde{Z}_1, & X_l &:= \tilde{X}_1 Z_2, \\
\hat{Z}_2 &:= \tilde{Z}_2, & \hat{X}_2 &:= \tilde{X}_2 (Z_2^\dagger)^2 Z_3, \\
\hat{Z}_i &:= \tilde{Z}_i, & \hat{X}_i &:= \tilde{X}_i Z_{i-1} (Z_i^\dagger)^2 Z_{i+1}, \quad i \geq 3.
\end{aligned}
\tag{3.14}
$$

After renaming $\hat{X}_i \to X_i, \hat{Z}_i \to Z_i, i \geq 2$, the half-gauged Hamiltonian becomes

$$
\begin{aligned}
H_{\text{half-gauged}} = & - g^{-1} \sum_{i \leq 0} Z_{i-1} (Z_i^\dagger)^2 Z_{i+1} - g \sum_{i \leq 0} X_i \\
& - g^{-1} Z_0 (Z_1^\dagger)^2 X_l - g X_1 - g^{-1} Z_1 X_2 - g Z_l (Z_2^\dagger)^2 Z_3 \\
& - g^{-1} \sum_{i \geq 3} X_i - g \sum_{i \geq 3} Z_{i-1} (Z_i^\dagger)^2 Z_{i+1} + (\text{h.c.}).
\end{aligned}
\tag{3.15}
$$

There is an interface located at the link (1,2) separating two theories which are exchanged by inverting the coupling $g \to g^{-1}$. By inserting the duality interface, we couple the system with one ancillary qudit $|s_l\rangle$ with Pauli operators $X_l, Z_l$ acting on this qudit.

Under the following local unitary transformation

$$
CZ_{2,3} (CZ_{2,l}^\dagger)^2 (U_2^H)^\dagger S_{2,l} : \begin{cases}
X_l \to Z_2, & Z_l \to X_2^\dagger Z_l^2 Z_3^\dagger, \\
X_2 \to X_l (Z_2^\dagger)^2, & Z_2 \to Z_l, \\
X_3 \to X_3 Z_2,
\end{cases}
\tag{3.16}
$$

the half-gauged Hamiltonian becomes

$$
\begin{aligned}
H_{\text{half-gauged}} \to & - g^{-1} \sum_{i \leq 1} Z_{i-1} (Z_i^\dagger)^2 Z_{i+1} - g \sum_{i \leq 0} X_i \\
& - g^{-1} Z_1 (Z_2^\dagger)^2 X_l - g X_2 - g^{-1} Z_2 X_3 - g Z_l (Z_3^\dagger)^2 Z_4 \\
& - g^{-1} \sum_{i \geq 4} X_i - g \sum_{i \geq 3} Z_{i-1} (Z_i^\dagger)^2 Z_{i+1} + (\text{h.c.}),
\end{aligned}
\tag{3.17}
$$

where the interface is moved to the link (2,3) as shown in Fig. 2. This duality interface is therefore *topological*. In (3.16), $CZ_{i,j}$ is the control-$Z$ gate, $U_j^H$ is the Hadamard gate and $S_{i,j}$ is the swap gate. Their definition and transformation on local operators can be found in Table. 2.

**Dipole KW duality defect**

When the coupling $g = 1$, the duality interface becomes a duality defect. The defect Hamiltonian with the dipole KW duality defect $\hat{\mathsf{D}}$ at link $(i_0, i_0 + 1)$ is

$$
H_{\hat{\mathsf{D}}}^{(i_0, i_0+1)} = - \sum_{i \neq i_0, i_0+1} (Z_{i-1}(Z_i^\dagger)^2 Z_{i+1} + X_i)
$$
$$
- (Z_{i_0-1}(Z_{i_0}^\dagger)^2 X_l + X_{i_0} + Z_l(Z_{i_0+1}^\dagger)^2 Z_{i_0+2} + Z_{i_0} X_{i_0+1}) + \text{(h.c.)}.
\tag{3.18}
$$

This defect is topological and can be moved to link $(i_0 + 1, i_0 + 2)$ by a movement operator

$$
U_{\hat{\mathsf{D}}}^{i_0+1} := CZ_{i_0+1,i_0+2}(CZ_{i_0+1,l}^\dagger)^2 (U_{i_0+1}^H)^\dagger S_{i_0+1,l}.
\tag{3.19}
$$

Since the defect is topological, we can insert two defects far away from each other, move them together and fuse them. Without loss of generality, we can move the two dipole KW defects to links $(1, 2)$ and $(2, 3)$

$$
H_{\hat{\mathsf{D}};\hat{\mathsf{D}}}^{(1,2),(2,3)} = - \sum_{i \neq 1,2,3} (Z_{i-1}(Z_i^\dagger)^2 Z_{i+1} + X_i)
$$
$$
- (Z_0(Z_1^\dagger)^2 X_{l_1} + X_1 + Z_{l_1}(Z_2^\dagger)^2 X_{l_2} + Z_1 X_2 + Z_{l_2}(Z_3^\dagger)^2 Z_4 + Z_2 X_3) + \text{(h.c.)},
\tag{3.20}
$$

where $X_{l_1}, Z_{l_1}, X_{l_2}, Z_{l_2}$ act on the ancillary qudit $|s_{l_1}\rangle, |s_{l_2}\rangle$ associated with the two defects. We need to apply the fusion operator

$$
\lambda_{\hat{\mathsf{D}} \otimes \hat{\mathsf{D}}}^{(1,2,3)} = CZ_{3,l_2}^\dagger CZ_{2,l_1}^\dagger (CZ_{2,l_2})^2 S_{2,l_2} : \begin{cases} X_{l_1} \rightarrow X_{l_1} Z_2^\dagger, \\ X_{l_2} \rightarrow X_2 Z_{l_2}^2 Z_{l_1}^\dagger, & Z_{l_2} \rightarrow Z_2, \\ X_2 \rightarrow X_{l_2} Z_2^2 Z_3^\dagger, & Z_2 \rightarrow Z_{l_2}, \\ X_3 \rightarrow X_3 Z_{l_2}^\dagger, \end{cases}
\tag{3.21}
$$

and the defect Hamiltonian after fusion is

$$
H_{\hat{\mathsf{D}};\hat{\mathsf{D}}}^{(1,2),(2,3)} \rightarrow H_{\hat{\mathsf{D}} \otimes \hat{\mathsf{D}}}^{(1,2)} = - \sum_{i \neq 1,2} (Z_{i-1}(Z_i^\dagger)^2 Z_{i+1} + X_i)
$$
$$
- (Z_0(Z_1^\dagger)^2 Z_2^\dagger X_{l_1} + X_1 + Z_1^\dagger(Z_2^\dagger)^2 Z_3 X_{l_2}^\dagger + X_2) + \text{(h.c.)}.
\tag{3.22}
$$

This Hamiltonian has two decoupled degrees of freedom on ancillary sites $l_1$ and $l_2$; $X_{l_1}, X_{l_2}$ become symmetry operators of the new Hamiltonian whose Hilbert space is decomposed into a direct sum of the eigenspaces of $X_1$ and $X_2$. Compared with the defect Hamiltonian with insertion of invertible topological defects of $\mathsf{C}, \eta_Q, \eta_D$ in Appendix B, one finds that after picking the eigenvalue of $X_1$ and $X_2$, the fused defect Hamiltonian corresponds to different channels of the fusion rule

$$
\hat{\mathsf{D}} \times \hat{\mathsf{D}} = \left( \sum_{k=1}^N \eta_Q^k \right) \left( \sum_{k=1}^N \eta_D^k \right) \mathsf{C}.
\tag{3.23}
$$

## 3.3 Bilinear phase map representation and symmetry-twist sectors

The dipole KW transformation has the following BMP representation under periodic boundary conditions

$$\hat{\mathsf{D}} = \sum_{\{s_j\},\{s'_j\}} \omega^{-\sum_{i=1}^{L}(s'_{i+1}+s'_{i-1}-2s'_i)s_i} \left|\{s'_j\}\right\rangle \left\langle\{s_j\}\right|$$
$$= \sum_{\{s_j\},\{s'_j\}} \omega^{-\sum_{i=1}^{L}(s_{i+1}+s_{i-1}-2s_i)s'_i} \left|\{s'_j\}\right\rangle \left\langle\{s_j\}\right|. \tag{3.24}$$

where the exponent respects the dipole interaction and the minimal gauge coupling. While the fusion algebras have been studied from decomposition into the seed transformation in Sec. 2, here we instead use the BPM representation to study the mapping of symmetry-twist sectors before and after the dipole KW transformation. This will be useful in the analysis of the anomaly of the dipole KW symmetry in the next section.

**Mapping between symmetry-twist sectors**

Since there are charge and dipole symmetries, the states can be organized into eigenstates of $\eta_Q$ and $\eta_D$ with symmetry eigenvalue $\omega^{u_Q}$ and $\omega^{u_D}$ where $u_Q, u_D \in \mathbb{Z}_N$. Moreover, one can also use the $\mathbb{Z}_N^Q$ and $\mathbb{Z}_N^D$ symmetries to twist the boundary condition of the spins as

$$s_{i+L} := s_i + t_Q + t_D i, \quad |s_{i+L}\rangle_{i+L} := |s_i + t_Q + t_D i\rangle_i, \tag{3.25}$$

with $t_Q, t_D \in \mathbb{Z}_N$. The Pauli $Z$ operators also have a boundary condition:

$$Z_{i+L} = \omega^{t_Q+t_D i} Z_i. \tag{3.26}$$

One can organize the Hilbert space into $N^4$ symmetry-twist sectors, labeled by $(u_Q, t_Q, u_D, t_D)$. Similarly, one can also label the symmetry-twist sectors of spins after the dipole KW transformation by $(\widehat{u}_Q, \widehat{t}_Q, \widehat{u}_D, \widehat{t}_D)$. We would like to find out the relation between the sectors before and after the dipole KW transformation.

The BMP representation is modified with general boundary conditions

$$\hat{\mathsf{D}} = \sum_{\{s_j\},\{s'_j\}} \omega^{\sum_{i=1}^{L} -(s'_{i+1}+s'_{i-1}-2s'_i)s_i-(t_Q+t_D)s'_L+t_Q s'_1} \left|\{s'_j\}\right\rangle \left\langle\{s_j\}\right|$$
$$= \sum_{\{s_j\},\{s'_j\}} \omega^{\sum_{i=1}^{L} -(s_{i+1}+s_{i-1}-2s_i)s'_i-(\widehat{t}_Q+\widehat{t}_D)s_L+\widehat{t}_Q s_1} \left|\{s'_j\}\right\rangle \left\langle\{s_j\}\right|. \tag{3.27}$$

Let us first consider $\hat{\mathsf{D}} \times \eta_Q$ acting on an arbitrary state $|\psi\rangle = \sum_{\{s_i\}} \psi_{\{s_i\}} |\{s_i\}\rangle$.

$$
\begin{aligned}
\hat{\mathsf{D}} \times \eta_Q |\psi\rangle &= \hat{\mathsf{D}} \sum_{\{s_i\}} \psi_{\{s_i\}} |\{s_i + 1\}\rangle \\
&= \sum_{\{s_i'\},\{s_i\}} \psi_{\{s_i\}} \omega^{-\sum_{i=1}^{L}(s_{i+1}+s_{i-1}-2s_i)s_i' - (\widehat{t}_Q + \widehat{t}_D)(s_L+1) + \widehat{t}_Q(s_1+1)} |\{s_i'\}\rangle \\
&= \omega^{-\widehat{t}_D} \sum_{\{s_i'\},\{s_i\}} \psi_{\{s_i\}} \omega^{\sum_{i=1}^{L} -(s_{i+1}+s_{i-1}-2s_i)s_i' - (\widehat{t}_Q+\widehat{t}_D)s_L + \widehat{t}_Q s_1} |\{s_i'\}\rangle \\
&= \omega^{-\widehat{t}_D} \hat{\mathsf{D}} |\psi\rangle .
\end{aligned}
\tag{3.28}
$$

The result implies that for any eigenstate $|\Psi\rangle$ with

$$
\eta_Q |\Psi\rangle = \omega^{u_Q} |\Psi\rangle ,
\tag{3.29}
$$

we have

$$
\omega^{-\widehat{t}_D} \hat{\mathsf{D}} |\Psi\rangle = \hat{\mathsf{D}} \times \eta_Q |\Psi\rangle = \omega^{u_Q} \hat{\mathsf{D}} |\Psi\rangle ,
\tag{3.30}
$$

namely

$$
\widehat{t}_D = -u_Q.
\tag{3.31}
$$

On the other hand, one can also consider $\widehat{\eta}_Q \times \hat{\mathsf{D}}$ acting on a arbitrary state $|\psi\rangle$,

$$
\begin{aligned}
\widehat{\eta}_Q \times \hat{\mathsf{D}} |\psi\rangle &= \widehat{\eta}_Q \sum_{\{s_i'\},\{s_i\}} \psi_{\{s_i\}} \omega^{\sum_{i=1}^{L} -(s_{i+1}'+s_{i-1}'-2s_i')s_i - (t_Q+t_D)s_L' + t_Q s_1'} |\{s_i'\}\rangle \\
&= \sum_{\{s_i'\},\{s_i\}} \psi_{\{s_i\}} \omega^{\sum_{i=1}^{L} -(s_{i+1}'+s_{i-1}'-2s_i')s_i - (t_Q+t_D)s_L' + t_Q s_1'} |\{s_i' + 1\}\rangle \\
&= \omega^{t_D} \hat{\mathsf{D}} |\psi\rangle .
\end{aligned}
\tag{3.32}
$$

This shows that the dual $\mathbb{Z}_N^Q$-symmetry charge sector after dipole KW transformation is identified with the twisted sector before this transformation,

$$
\widehat{u}_Q = t_D.
\tag{3.33}
$$

By similar calculations, one can determine the correspondence between the remaining sectors:

$$
\widehat{t}_Q = u_D, \quad \widehat{u}_D = -t_Q.
\tag{3.34}
$$

In summary, we have the following map of the symmetry-twist sectors:

$$
(\widehat{u}_Q, \widehat{t}_Q, \widehat{u}_D, \widehat{t}_D) = (t_D, u_D, -t_Q, -u_Q).
\tag{3.35}
$$

# 4  Anomaly of dipole Kramers-Wannier symmetry

In this section, we discuss the 't Hooft anomaly of the dipole KW symmetry by checking whether there is a dipole SPT phase invariant under such symmetry. A theory with an anomaly-free symmetry in the low energy should be uniquely gapped. Therefore, we will find an anomaly if we cannot find a gapped phase with a unique ground state that is invariant under the dipole KW transformation with a certain $N$. The anomaly imposes a nonperturbative constraint on the self-dual theories that they must have continuous or first-order phase transitions. This can help us understand, for example, the phase diagrams of systems with $\mathbb{Z}_3$ dipole symmetry.

## 4.1  Anomaly free condition for general $N$

Suppose we have a theory with both invertible $\mathbb{Z}_N^Q \times \mathbb{Z}_N^D$ symmetry and the non-invertible dipole KW symmetry. We will prove the anomaly of the dipole KW symmetry by contradiction. Because the charge and dipole symmetry $\mathbb{Z}_N^Q \times \mathbb{Z}_N^D$ is anomaly free, the symmetric theory is compatible with a gapped phase with one ground state, i.e. dipole SPT phase. We exclude the trivial phase because it is mapped to a dipole SSB phase under the dipole KW transformation and therefore does not have a dipole KW symmetry. The dipole SPT phase has been studied [130] and classified [129], which is given by an element of $H^2(\mathbb{Z}_N \times \mathbb{Z}_N, U(1))/H^2(\mathbb{Z}_N, U(1))^2 = \mathbb{Z}_N$. A simple example is the stabilizer Hamiltonian:

$$H_{\text{SPT-}k} = -\sum_{i=1}^{L}\sum_{m=1}^{N}[(Z_{i-1}Z_i^\dagger)^k X_i (Z_i^\dagger Z_{i+1})^k]^m, \tag{4.1}$$

where level $k \in \mathbb{Z}_N$ corresponds to different classes. The SPT Hamiltonian is constructed from the trivial-phase Hamiltonian

$$H_{\text{triv}} = -\sum_{i=1}^{L}\sum_{m=1}^{N}(X_i)^m, \tag{4.2}$$

by decorated domain wall construction

$$H_{\text{SPT-}k} = T_D^k H_{\text{triv}} T_D^{-k}, \quad T_D = \prod_{i=1}^{L} CZ_{i-1,i} CZ_{i,i}^\dagger, \tag{4.3}$$

where the transformation of control-$Z$ gate on Pauli operators can be found in Table 2.

Since all the terms in the SPT Hamiltonian (4.1) commute with each other, the ground state satisfies

$$(Z_{i-1}Z_i^\dagger)^k X_i (Z_i^\dagger Z_{i+1})^k |\text{G.S.}\rangle = |\text{G.S.}\rangle. \tag{4.4}$$

The SPT phases with different $k$ are characterized by the charges of ground state $|\text{G.S.}\rangle_{t_Q, t_D}$ with twisted boundary conditions labelled by $(t_Q, t_D)$ [120, 131–133]. It is straightforward to calculate

the charges of symmetry operator $\eta_Q, \eta_D$ on ground state $|\text{G.S.}\rangle_{t_Q,t_D}$

$$(\prod_{i=1}^{L} X_i)\,|\text{G.S.}\rangle_{t_Q,t_D} = \prod_{i=1}^{L}(Z_{i-1}^\dagger Z_i^2 Z_{i+1}^\dagger)^k\,|\text{G.S.}\rangle_{t_Q,t_D} = (Z_0^\dagger Z_1 Z_L Z_{L+1}^\dagger)^k\,|\text{G.S.}\rangle_{t_Q,t_D} = \omega^{-kt_D}\,|\text{G.S.}\rangle_{t_Q,t_D},$$

$$(\prod_{i=1}^{L} X_i^i)\,|\text{G.S.}\rangle_{t_Q,t_D} = \prod_{i=1}^{L}(Z_{i-1}^\dagger Z_i^2 Z_{i+1}^\dagger)^{ki}\,|\text{G.S.}\rangle_{t_Q,t_D} = (Z_0^\dagger Z_L)^k\,|\text{G.S.}\rangle_{t_Q,t_D} = \omega^{kt_Q}\,|\text{G.S.}\rangle_{t_Q,t_D},$$

$$(4.5)$$

where we used the twist boundary condition of Pauli operators in the last equality in each equation. Therefore the ground state $|\text{G.S.}\rangle_{t_Q,t_D}$ is in the symmetry-twist sector labelled by

$$(u_Q = -kt_D, t_Q, u_D = kt_Q, t_D). \tag{4.6}$$

If the ground states after the dipole KW transformation do not stay in the same sector, the SPT phase is not invariant under this transformation. If for given $N$ every $k$ we cannot find an SPT that is invariant under the dipole KW transformation, then the symmetry is anomalous.

Now let us check whether there is an SPT phase invariant under dipole KW transformation. The dual Hamiltonian is

$$H_k' = -\sum_{i=1}^{L}\sum_{m=1}^{N}[Z_{i-1}^\dagger Z_i X_i^k Z_i Z_{i+1}^\dagger]^m. \tag{4.7}$$

The dual Hamiltonian is not necessarily an SPT but still a stabilizer model. We will classify the dual Hamiltonian with different $k$, $N$ into different gapped phases in Sec. 5.2. The corresponding ground state(s) $|\text{G.S.}'\rangle$ of $H_k'$ satisfies

$$Z_{i-1}^\dagger Z_i X_i^k Z_i Z_{i+1}^\dagger\,|\text{G.S.}'\rangle = |\text{G.S.}'\rangle. \tag{4.8}$$

In the dual systems, we label twist boundary conditions of dual systems using $\widehat{t}_Q, \widehat{t}_D$. Then the charges of ground state $|\text{G.S.}'\rangle_{\widehat{t}_Q,\widehat{t}_D}$ with twisted boundary conditions are

$$(\prod_{i=1}^{L} X_i)^k\,|\text{G.S.}'\rangle_{\widehat{t}_Q,\widehat{t}_D} = \prod_{i=1}^{L}(Z_{i-1}^\dagger Z_i^2 Z_{i+1}^\dagger)^{-1}\,|\text{G.S.}'\rangle_{\widehat{t}_Q,\widehat{t}_D} = (Z_0^\dagger Z_1 Z_L Z_{L+1}^\dagger)^{-1}\,|\text{G.S.}'\rangle_{\widehat{t}_Q,\widehat{t}_D} = \omega^{\widehat{t}_D}\,|\text{G.S.}'\rangle_{\widehat{t}_Q,\widehat{t}_D},$$

$$(\prod_{i=1}^{L} X_i^i)^k\,|\text{G.S.}'\rangle_{\widehat{t}_Q,\widehat{t}_D} = \prod_{i=1}^{L}(Z_{i-1}^\dagger Z_i^2 Z_{i+1}^\dagger)^{-i}\,|\text{G.S.}'\rangle_{\widehat{t}_Q,\widehat{t}_D} = (Z_0^\dagger Z_L)^{-1}\,|\text{G.S.}'\rangle_{\widehat{t}_Q,\widehat{t}_D} = \omega^{-\widehat{t}_Q}\,|\text{G.S.}'\rangle_{\widehat{t}_Q,\widehat{t}_D},$$

$$(4.9)$$

which shows that the ground state $|\text{G.S.}'\rangle_{\widehat{t}_Q,\widehat{t}_D}$ is in the symmetry-twist sector labelled by

$$(\widehat{u}_Q, \widehat{t}_Q = -k\widehat{u}_D, \widehat{u}_D, \widehat{t}_D = k\widehat{u}_Q). \tag{4.10}$$

Therefore, if there is an SPT phase invariant under dipole KW transformation, the ground state charges of (4.7) and (4.1) under the same twisted boundary condition should be consistent with each other, that is

$$\widehat{u}_Q \overset{(4.6)}{=} -k\widehat{t}_D \overset{(4.10)}{=} -k^2\widehat{u}_Q, \quad \widehat{u}_D \overset{(4.6)}{=} k\widehat{t}_Q \overset{(4.10)}{=} -k^2\widehat{u}_D, \tag{4.11}$$

where all equations are valued modulo $N$. Thus $k$ should satisfy $k^2 = -1 \bmod N$, i.e., $-1$ is a quadratic residue modulo $N$, which is the necessary anomaly-free condition for dipole KW symmetry.[10]

Indeed, this condition is also a sufficient anomaly-free condition. This is because the dual Hamiltonian (4.7) is the same as the Hamiltonian (4.1) in this case. When $k^2 = -1 \bmod N$, $N$ is coprime with $k$. Hence, for each $m \in \mathbb{Z}_N$, there exist a unique $j_m \in \mathbb{Z}_N$ satisfying $kj_m = m \bmod N$. As a result, we have the

$$
\begin{aligned}
H_k' &= -\sum_{i=1}^{L}\sum_{m=1}^{N}[Z_{i-1}^\dagger Z_i X_i^k Z_i Z_{i+1}^\dagger]^{j_m} \\
&= -\sum_{i=1}^{L}\sum_{m=1}^{N}(Z_{i-1}^\dagger Z_i)^{j_m} X_i^m (Z_i Z_{i+1}^\dagger)^{j_m} = H_{\text{SPT-}k},
\end{aligned}
\tag{4.12}
$$

where in the last equation, we use the fact that $j_m = -k^2 j_m = -km \bmod N$.

In summary, when $-1$ is a quadratic residue modulo $N$, the dipole KW symmetry is anomaly-free and there exists a dipole SPT also protected by this non-invertible symmetry; otherwise it is an anomalous symmetry.

## 4.2   Anomaly and phase diagram for $N = 3$

When $N = 3$, the dipole KW symmetry is anomalous because $-1$ is not a quadratic residue modulo 3. Indeed, this anomaly can help us understand the phase transition in the phase diagram of the dipole Ising model:

$$
\begin{aligned}
H_{\text{dipole-Ising}} &= -g^{-1}\sum_{i=1}^{L} Z_{i-1}(Z_i^\dagger)^2 Z_{i+1} - g\sum_{i=1}^{L} X_i + (\text{h.c.}) \\
&= \sqrt{g^2 + g^{-2}}\sum_{i=1}^{L}(\cos\theta Z_{i-1}(Z_i^\dagger)^2 Z_{i+1} + \sin\theta X_i) + (\text{h.c.}),
\end{aligned}
\tag{4.13}
$$

where we defined an angle $\theta$ by $\cos\theta = -g^{-1}/\sqrt{g^2 + g^{-2}}, \sin\theta = -g/\sqrt{g^2 + g^{-2}}$. The dipole Ising model can be mapped to the XZ chain

$$H_{\text{XZ}} = \sqrt{g^2 + g^{-2}}\sum_{i=1}^{L}(\cos\theta X_{i-1}^\dagger X_i + \sin\theta Z_{i-1}^\dagger Z_i) + (\text{h.c.}), \tag{4.14}$$

---

[10]Similar constraints also appear in the anomaly of non-invertible duality defects in $(3+1)d$ [19].

by gauging the ordinary $\mathbb{Z}_3^Q$ symmetry. The $\mathbb{Z}_3^D$ dipole symmetry becomes a $\mathbb{Z}_3^Z$ symmetry generated by $\prod_{i=1}^L Z_i$ and we have a new quantum $\mathbb{Z}_N^X$ symmetry generated by $\prod_{i=1}^L X_i$.

The phase diagram of Hamiltonian (4.14) has been determined by numerical calculation in [109, 110], under the name "quantum torus chain". We show the phase diagram in the left part of Fig. 3, which can be summarized as follows:

1. The phase diagram exhibits a symmetric pattern across the line of $\theta = 0.25\pi$, $\theta = 1.25\pi$, at which points the model is invariant under the global Hadamard gate $U^H : X_i \to Z_i^\dagger, Z_i \to X_i, \forall i$.

2. Around $\theta = 0.25\pi$, this model hosts a large gapless phase region, $\theta \in (-0.1\pi, 0) \cup (0, 0.5\pi) \cup (0.5\pi, 0.6\pi)$, with center charge $c = 2$. At two particular points $\theta = 0, 0.5\pi$ there are first-order phase transitions with exponentially large ground state degeneracy.

3. There is a first order phase transition at $\theta = 1.25\pi$, which separates the $\mathbb{Z}_3^Z$ SSB phase with nonzero $\langle X_i \rangle$ when $\theta \in (0.6\pi, 1.25\pi)$ and the $\mathbb{Z}_3^X$ SSB phase with nonzero $\langle Z_i \rangle$ when $\theta \in (1.25\pi, 1.9\pi)$. The first-order phase transitions at $\theta = 0.6\pi, 1.9\pi$ separate gapped SSB phases and gapless phases.

As the KW transformation preserves the structure of the phase diagram and the center charge of gapless region, we can obtain the phase diagram of dipole Ising model. We show the phase diagram in the right part of Fig. 3 and summarize it here

1. The phase diagram exhibits a symmetric pattern across the line of $\theta = 0.25\pi$, $\theta = 1.25\pi$ at which points the model is invariant under the dipole KW transformation $\hat{D}$.

2. Around $\theta = 0.25\pi$, the system is in the gapless phases with $c = 2$ when $\theta \in (-0.1\pi, 0) \cup (0, 0.5\pi) \cup (0.5\pi, 0.6\pi)$. These phases are separated by first-order phase transitions at $\theta = 0, 0.5\pi$, where the Hamiltonian is dominated by the term $Z_{i-1}(Z_i^\dagger)^2 Z_{i+1} + $ (h.c.) and $X_i + $ (h.c.) respectively. therefore, there is also an exponentially large ground state degeneracy $N_{GS} \sim 2^L$.

3. The $\theta = 1.25\pi$ is a first order phase transition separating a trivial gapped phase and a dipole SSB phase. When $\theta \in (1.25\pi, 1.9\pi)$, the Hamiltonian is dominated by $-X_i + $ (h.c.) and the system is in the trivially gapped phase. When $\theta \in (0.6\pi, 1.25\pi)$, the Hamiltonian is dominated by $-Z_{i-1}(Z_i^\dagger)^2 Z_{i+1} + $ (h.c.). There are nine ground states when $L \to \infty$ (with the sequence of $L \equiv 0$ mod 3) with nonzero $\langle Z_{i-1}^\dagger Z_i \rangle$ and $\langle Z_i \rangle$ and the system is in the $\mathbb{Z}_3^Q \times \mathbb{Z}_3^D$ SSB phase. There are also first-order phase transitions at $\theta = 0.6\pi, 1.9\pi$ between different gapped phases and gapless phases.

4. We highlight that the gapless feature of self-dual points at $\theta = 0.25\pi$ ($g = -1$) and first-order phase transition at self-dual points $1.25\pi$ ($g = 1$) are consistent with the anomalous dipole KW symmetry because the anomaly forbids the system to be uniquely gapped.

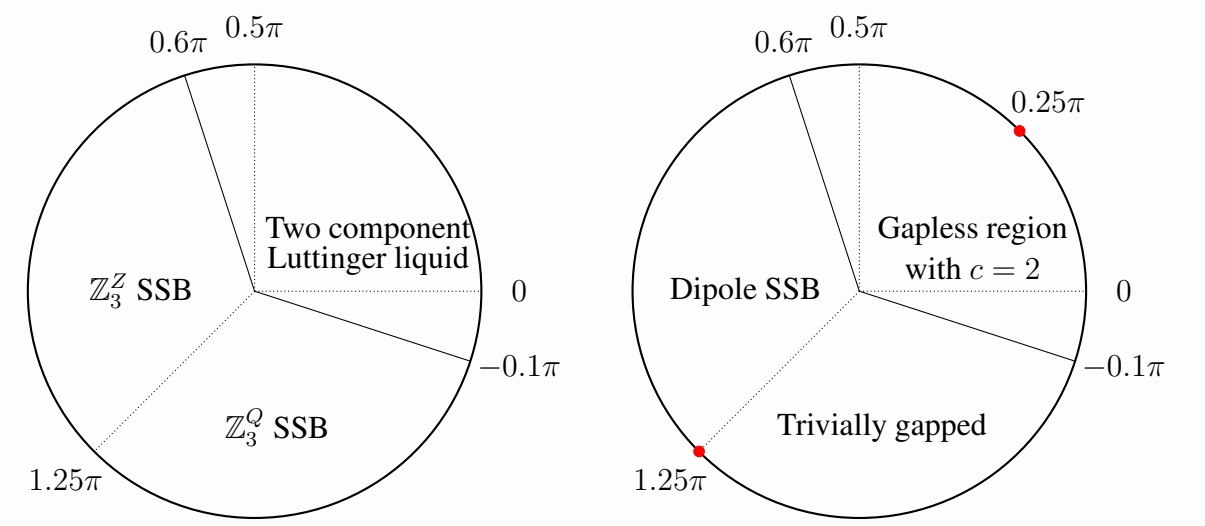

Figure 3: The phase diagram of XZ model (4.14) (left) and the dipole Ising model (right) where dashed lines are first order phases transition and solid lines are continuous phase transitions. The red dots $\theta = 0.25\pi, 1.25\pi$ (right) are critical points where the dipole Ising model has the anomalous non-invertible symmetry.

# 5 Generalized Kennedy-Tasaki transformation associated with $\mathbb{Z}_N^Q \times \mathbb{Z}_N^D$ symmetry

In this section, we will discuss generalized Kennedy-Tasaki (KT) transformations associated with $\mathbb{Z}_N^Q \times \mathbb{Z}_N^D$ symmetry [119], which relate dipole SSB phases and dipole SPT phases.

## 5.1 Construction of KT transformations

In Sec 4.1, we defined a $T_D$ transformation for systems with dipole symmetries

$$T_D := \prod_{i=1}^{L} CZ_{i-1,i} CZ_{i,i}^{\dagger}, \tag{5.1}$$

which generates the dipole SPT phases from the trivial phase:

$$T_D^k : \quad H_{\text{triv}} \to H_{\text{SPT-}k}. \tag{5.2}$$

A single $T_D$ transformation will increase the SPT level by one[11]

$$T_D : \quad H_{\text{SPT-}k} \to H_{\text{SPT-}(k+1)}. \tag{5.3}$$

---

[11]Here we use the notation that the trivial phase is the SPT phase with level 0.

$T_D$ is an invertible transformation and $T_D^{-k}$ maps a dipole SPT with level $k$ to a trivial phase (with level $0$). Because we can map a trivial phase to the dipole SSB phase

$$H_{\text{SSB}} = -\sum_{i=1}^{L}\sum_{m=1}^{N}(Z_{i-1}^{\dagger}Z_i^2 Z_{i+1}^{\dagger})^m, \tag{5.4}$$

via the dipole KW transformation $\hat{\mathsf{D}}$, and further the SSB phase is invariant under the $T_D$ transformation, we can define a duality transformation, the generalized KT transformation, between the SSB phase and the SPT phase with each $k$

$$\mathsf{KT}_k := T_D^k\hat{\mathsf{D}}T_D^{-k} : \quad H_{\text{SPT-}k} \to H_{\text{SSB}}, \quad H_{\text{SSB}} \to H_{\text{SPT-}k}. \tag{5.5}$$

The duality web is shown in Fig. 4.

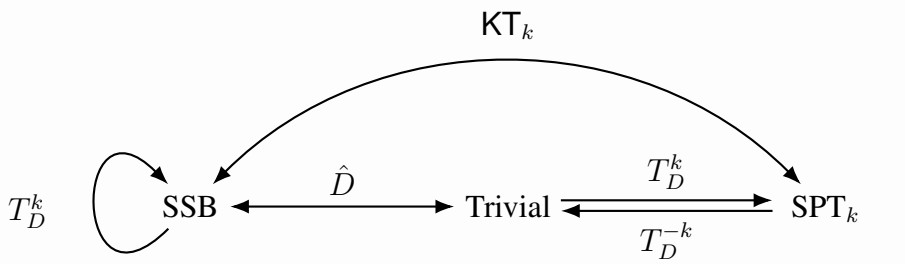

Figure 4: Transformations between the trivial phase, the SSB phase and the SPT phase with level $k$. KT transformation $T_D^k\hat{\mathsf{D}}T_D^{-k}$ is a duality map between the SSB phase and SPT phase with level $k$.

This KT transformation is non-invertible with the fusion rule

$$\mathsf{KT}_k \times \mathsf{KT}_k = \left(\sum_{m=1}^{N}\eta_Q^m\right)\left(\sum_{m=1}^{N}\eta_D^m\right)\mathsf{C}. \tag{5.6}$$

Thus when acting $\mathsf{KT}_k$ twice, the Hamiltonian with $\mathbb{Z}_N^Q \times \mathbb{Z}_N^D$ and charge conjugation symmetry is invariant, which implies $\mathsf{KT}_k$ is indeed a duality transformation for such Hamiltonian, e.g. the SPT Hamiltonian (4.1).[12]

## 5.2 Mapping of gapped phases under KT transformation

In the previous subsection, we define the KT duality transformation $\mathsf{KT}_k$ for each dipole SPT with level $k$. Now, we would like to study the general transformation of $\mathsf{KT}_p$ on SPT with a different

---

[12]Indeed, if we apply charge conjugation to the SPT phase and SSB phase, one can show the features of these phases, e.g., ground state charge under twisted boundaries and ground state degeneracy, are kept invariant.

level $p + k$ with $k \neq 0$. We can calculate the transformation in sequence because of the definition $\mathsf{KT}_p = T_D^p \hat{\mathsf{D}} T_D^{-p}$. In the first step, $T_D^{-p}$ will map the SPT with level $p + k$ to the SPT with level $k$.

The second step is to determine which gapped phase the SPT will become after the dipole KW transformation, which depends on whether $k$ and $N$ are coprime. Then in the third step, $T_D^{-p}$ will further stack some SPTs to the model. Suppose after the KT transformation $\mathsf{KT}_p$ on $H_{\text{SPT-}(p+k)}$ we get a dual Hamiltonian. We summarize the results here and leave the derivation later. The duality web is shown in Fig. 5.

1. When $\gcd(k, N) > 1$, the dual Hamiltonian is a partial SSB phase with unbroken $\mathbb{Z}_{N/\gcd(k,N)}^Q \times \mathbb{Z}_{N/\gcd(k,N)}^D$ symmetry and stacked with a SPT phase of unbroken symmetry with level $n + p \times \gcd(k, N)$, where $nk = -\gcd(k, N) \bmod N$.

2. When $\gcd(k, N) = 1$, the dual Hamiltonian is a SPT phase with level $n' + p$, where $n'k = -1 \bmod N$.

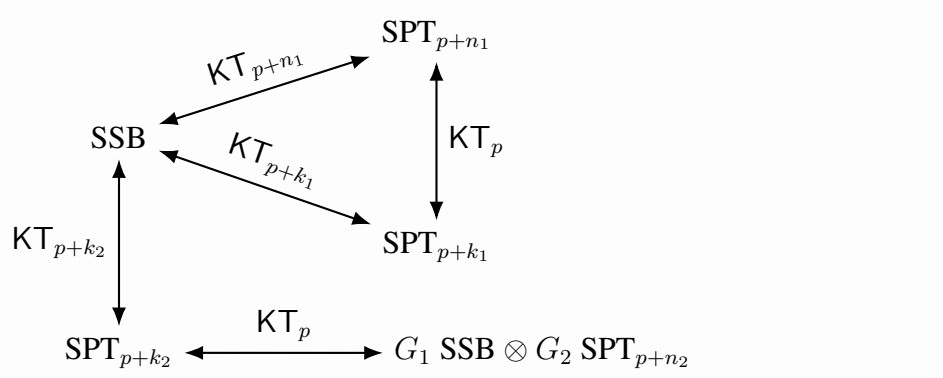

Figure 5: Duality web between various gapped phases. Unless specified, the SSB phase breaks all $G = \mathbb{Z}_N^Q \times \mathbb{Z}_N^D$ symmetry and the SPT phase is protected $G$ symmetry. $k_1, n_1$ are coprime with $N$ and $n_1 k_1 = -1 \bmod N$. $k_2$ is not coprime with $N$ and $k_2 n_2 = -\gcd(k_2, N) \bmod N$. $G_1 = \mathbb{Z}_{N/\gcd(k_2,N)}^Q \times \mathbb{Z}_{N/\gcd(k_2,N)}^D$ and $G_2 = \mathbb{Z}_{\gcd(k_2,N)}^Q \times \mathbb{Z}_{\gcd(k_2,N)}^D$.

**Mapping of SPT phases under dipole KW transformation**

This part provides the technique details about the duality web of KT transformation on SPT with at a general level.

Start from the second step and suppose after the dipole KW transformation on $H_{\text{SPT-}k}$ we get the dual Hamiltonian $H_k'$. It is sufficient to study the ground-state degeneracy of the dual Hamiltonian with the periodic boundary condition. Recall the analysis in Sec. 4.1 and (4.10), where the ground states of the dual Hamiltonian are in symmetry-twist sector labeled by $(\widehat{u}_Q, \widehat{t}_Q = -k\widehat{u}_D, \widehat{u}_D, \widehat{t}_D = k\widehat{u}_Q)$. In the periodic boundary condition, we have

$$\widehat{t}_Q = -k\widehat{u}_D = 0, \quad \widehat{t}_D = k\widehat{u}_Q = 0, \bmod N. \tag{5.7}$$

Therefore, $\widehat{u}_D$ and $\widehat{u}_Q$ must be multiples of $N/\gcd(k,N)$ and the ground state degeneracy is $\gcd(k,N)^2$.

**When $\gcd(k,N) > 1$:** The dual Hamiltonian has degenerate ground states and the remaining unbroken subgroup $\mathbb{Z}^Q_{N/\gcd(k,N)} \times \mathbb{Z}^D_{N/\gcd(k,N)}$ is generated by $\eta_Q^{\gcd(k,N)}$ and $\eta_D^{\gcd(k,N)}$, as its ground state charge under PBC is trivial. The dual Hamiltonian is therefore in an SSB phase stacked with a $\mathbb{Z}^Q_{N/\gcd(k,N)} \times \mathbb{Z}^D_{N/\gcd(k,N)}$ SPT phase. Then we need to determine the level of the SPT phase.

Because of the broken symmetry, the twist sector label $\widehat{t}_Q, \widehat{t}_D$ takes value in $\gcd(k,N)\mathbb{Z}_{N/\gcd(k,N)}$. It is convenient to use the reduced twist variables

$$(\widehat{\mathsf{t}}_Q, \widehat{\mathsf{u}}_D) = (\widehat{t}_Q/\gcd(k,N), \widehat{t}_D/\gcd(k,N)) \in \mathbb{Z}^2_{N/\gcd(k,N)} \tag{5.8}$$

for twist sectors of the unbroken symmetry. Moreover, we can also define the symmetry sector of unbroken symmetry $(\widehat{\mathsf{u}}_Q, \widehat{\mathsf{u}}_D) = (\widehat{u}_Q, \widehat{u}_D) \bmod N/\gcd(k,N)$ where $(\widehat{\mathsf{u}}_Q, \widehat{\mathsf{u}}_D) \in \mathbb{Z}^2_{N/\gcd(k,N)}$. This is because any state $|\psi\rangle$ with $\mathbb{Z}^Q_N \times \mathbb{Z}^D_N$ charge $(\widehat{u}_Q, \widehat{u}_D)$ satisfies

$$\begin{aligned}
\eta_Q^{\gcd(k,N)}|\psi\rangle &= \omega^{\gcd(k,N)\widehat{u}_Q}|\psi\rangle = (\omega')^{\widehat{u}_Q}|\psi\rangle = (\omega')^{\widehat{\mathsf{u}}_Q}|\psi\rangle, \\
\eta_D^{\gcd(k,N)}|\psi\rangle &= \omega^{\gcd(k,N)\widehat{u}_D}|\psi\rangle = (\omega')^{\widehat{u}_D}|\psi\rangle = (\omega')^{\widehat{\mathsf{u}}_D}|\psi\rangle,
\end{aligned} \tag{5.9}$$

where $\eta_{Q(D)}^{\gcd(k,N)}$ is the generator of the unbroken symmetry and $\omega' = \omega^{\gcd(k,N)}$. To further determine which class the stacked SPT belongs to, we need to check the symmetry-twist sector of the ground states in terms of the unbroken symmetry $\mathbb{Z}^Q_{N/\gcd(k,N)} \times \mathbb{Z}^D_{N/\gcd(k,N)}$ to follow (4.6), i.e.

$$(\widehat{\mathsf{u}}_Q = -n\widehat{\mathsf{t}}_D, \widehat{\mathsf{t}}_Q, \widehat{\mathsf{u}}_D = n\widehat{\mathsf{t}}_Q) \bmod (N/\gcd(k,N)), \tag{5.10}$$

if the level is $n$. On the other hand, the symmetry-twist sector after the dipole KW transformation should satisfy (4.10) and we have

$$\widehat{t}_Q = -k\widehat{u}_D, \quad \widehat{t}_D = k\widehat{u}_Q \bmod N, \tag{5.11}$$

and

$$\widehat{\mathsf{t}}_Q = -(k/\gcd(k,N))\widehat{\mathsf{u}}_D, \quad \widehat{\mathsf{t}}_D = (k/\gcd(k,N))\widehat{\mathsf{u}}_Q \bmod (N/\gcd(k,N)), \tag{5.12}$$

in terms of the reduced variables. Since $N/\gcd(k,N)$ is coprime with $k/\gcd(k,N)$, we can find a unique integer $n \in \mathbb{Z}_{N/\gcd(k,N)}$ satisfying $n(k/\gcd(k,N)) = -1 \bmod (N/\gcd(k,N))$. Then we have

$$n\widehat{\mathsf{t}}_Q = -n(k/\gcd(k,N))\widehat{\mathsf{u}}_D = \widehat{\mathsf{u}}_D, \quad n\widehat{\mathsf{t}}_D = n(k/\gcd(k,N))\widehat{\mathsf{u}}_Q = -\widehat{\mathsf{u}}_Q \bmod (N/\gcd(k,N)). \tag{5.13}$$

Compared with (5.10), the stacked $\mathbb{Z}^Q_{N/\gcd(k,N)} \times \mathbb{Z}^D_{N/\gcd(k,N)}$ SPT has level $n$, where

$$n(k/\gcd(k,N)) = -1 \bmod (N/\gcd(k,N)) \quad \rightarrow \quad nk = -\gcd(k,N) \bmod N. \tag{5.14}$$

In the third step, $T_D^p$ transformation stacks a $\mathbb{Z}^Q_N \times \mathbb{Z}^D_N$ SPT phase with level $p$. To detect which $\mathbb{Z}^Q_{N/\gcd(k,N)} \times \mathbb{Z}^D_{N/\gcd(k,N)}$ class such an SPT belongs to, we consider a symmetry-twist sector of

unbroken $\mathbb{Z}^Q_{N/\gcd(k,N)} \times \mathbb{Z}^D_{N/\gcd(k,N)}$ symmetry labeled by $(\mathsf{u}_Q, \mathsf{t}_Q, \mathsf{u}_D, \mathsf{t}_D) \in \mathbb{Z}^4_{N/\gcd(k,N)}$, which has the relation with symmetry-twist sectors of $\mathbb{Z}^Q_N \times \mathbb{Z}^D_N$ symmetry:

$$(\mathsf{u}_Q, \mathsf{u}_D) = (u_Q, u_D) \bmod N/\gcd(k,N), \quad (t_Q, t_D) = (\gcd(k,N)\mathsf{t}_Q, \gcd(k,N)\mathsf{t}_D). \quad (5.15)$$

Recall the ground state of $\mathbb{Z}^Q_N \times \mathbb{Z}^D_N$ SPT with level $p$ is in the symmetry-twist sector labelled by

$$(u_Q = -p\, t_D, t_Q, u_D = p\, t_Q, t_D). \quad (5.16)$$

The ground state then also satisfies

$$(\mathsf{u}_Q = -p\, \gcd(k,N)\mathsf{t}_D, \mathsf{u}_D = p\, \gcd(k,N)\mathsf{t}_Q) \bmod N/\gcd(k,N), \quad (5.17)$$

which implies such SPT belongs to $\mathbb{Z}^Q_{N/\gcd(k,N)} \times \mathbb{Z}^D_{N/\gcd(k,N)}$ SPT phase with level $p \times \gcd(k,N)$.

Thus we can conclude that the KT-dual system is in a partial SSB phase with unbroken $\mathbb{Z}^Q_{N/\gcd(k,N)} \times \mathbb{Z}^D_{N/\gcd(k,N)}$ symmetry and stacked with an SPT phase of unbroken symmetry with level $n + p \times \gcd(k,N)$.

**When $\gcd(k,N) = 1$:** Because $k$ is coprime with $N$, the dual Hamiltonian is in an SPT phase. The symmetry-twist sector after the dipole KW transformation should satisfy (4.10)

$$\widehat{t}_Q = -k\widehat{u}_D, \quad \widehat{t}_D = k\widehat{u}_Q \bmod N. \quad (5.18)$$

Because $k$ is coprime with $N$, we can find an unique integer $n \in \mathbb{Z}_N$ such that $nk = -1 \bmod N$. therefore,

$$n\widehat{t}_Q = -nk\widehat{u}_D = \widehat{u}_D, \quad n\widehat{t}_D = nk\widehat{u}_Q = -\widehat{u}_Q \bmod N. \quad (5.19)$$

Compared with (4.6), the dual system is in the $\mathbb{Z}^Q_N \times \mathbb{Z}^D_N$ SPT phase with level $n$. Then in the third step, $T^p_D$ maps the SPT with level $n$ to an SPT with level $n + p$.

# 6 Conclusion and discussion

In this paper, we constructed the dipole KW transformation by composing the seed transformation and by gauging the charge and dipole symmetry. Then we study this new non-invertible symmetry through fusion algebras (2.28), topological defects (in Fig. 2) and the anomaly (and its constraints in the phase diagram in Fig. 3). As an application, we constructed generalized KT transformations (in Fig. 4) that connect various gapped phases with dipole symmetry in a duality web (in Fig. 5).

We studied the fusion algebra (2.28) of the dipole KW symmetry and other invertible symmetry operators. The fusion rule of non-invertible dipole KW symmetry mixes with the charge conjugation symmetry and therefore the whole fusion algebra is different from the $\mathrm{TY}(\mathbb{Z}_N \times \mathbb{Z}_N)$ fusion algebra. It is interesting to further determine the fusion category data, in particular the F-symbols which are related to the associativity of the fusion algebra and the anomaly of the fusion category symmetry.

We have shown that due to the anomaly of the non-invertible symmetry, the two self-dual points in the phase diagram of dipole Ising model with $N = 3$ cannot be uniquely gapped. This is consistent with numerical calculations in the anisotropic XZ model (which is the ordinary KW dual of the dipole Ising model), where one self-dual point is a first order phase transition and the other one is in a large gapless region with center charge $c = 2$. It will be interesting to investigate numerically whether the gapless phase will become unstable with symmetric perturbations when we choose $N$ such that the non-invertible symmetry is not anomalous.

It is also interesting to extend the composing construction. Consider a theory $A$ with a global symmetry generated by an operator $G$ and a non-invertible duality transformation $\mathsf{K}$ between theories $A$ and $B$. There is a non-invertible operator $\mathsf{K}^\dagger G \mathsf{K}$ that commutes with the Hamiltonian of the theory $B$. Although whether or not $\mathsf{K}^\dagger G \mathsf{K}$ is an interesting non-invertible symmetry depends on specific models, this might give a systematic construction and perhaps classification of certain types of lattice non-invertible symmetries. Here are examples in the literature with similar constructions. In $(2+1)d$, the subsystem non-invertible symmetry has been constructed [32] by composing the KW transformation acting on lines and columns. Recently in [134], the authors constructed a non-invertible symmetry in $(2+1)d$ with cosine function fusion rule using the sandwich construction.

In the end, we sketch another application inspired by [125]. In this paper, the authors studied the non-invertible symmetry in $(2+1)d$ lattice model with $\mathbb{Z}_2^{(0)} \times \mathbb{Z}_2^{(1)}$ symmetry. An example is the lattice $\mathbb{Z}_2$ gauge theory with the Ising matter

$$H = -\sum_v \prod_{l \ni v} \sigma_l^x - \sum_f \prod_{l \in f} \sigma_l^z - \sum_l \sigma_l^z - \sum_v X_v - \sum_{\langle v,v' \rangle} Z_v Z_{v'}, \tag{6.1}$$

where $\sigma_l^{x(z)}$ denotes a gauge spin on the link, $X_v, Z_v$ a matter spin on the site, and $\langle v, v' \rangle$ denote the link of nearest sites $v, v'$. The non-invertible symmetry is

$$\mathsf{D} = \frac{1}{2}\mathsf{SC} : \quad X_v \longleftrightarrow \prod_{l \ni v} \sigma_l^x, \quad Z_v Z_{v'} \longleftrightarrow \sigma_{\langle v,v' \rangle}^z, \tag{6.2}$$

where $\mathsf{S}$ is the non-invertible swap operator and $\mathsf{C}$ is the condensation operator whose precise definition can be found in [125]. We will take (6.2) as the seed transformation and construct a new non-invertible symmetry

$$\mathsf{D}U^H\mathsf{D} : \quad X_v \longleftrightarrow \prod_{l=\langle v,v' \rangle \ni v} Z_v Z_{v'} = \prod_{l=\langle v,v' \rangle \ni v} Z_{v'}, \tag{6.3}$$

where $U^H = \prod_l U_l^H$ is the product of Hadamard gate on every link. This non-invertible symmetry can be found in lattice models with subsystem $\mathbb{Z}_2$ symmetry acting on the diagonal lines. Especially, the $(2+1)d$ cluster model

$$H_{2d\,\text{cluster}} = -\sum_v X_v \prod_{l=\langle v,v' \rangle \ni v} Z_{v'}, \tag{6.4}$$

is protected by this non-invertible symmetry. We leave the detailed investigation of this non-invertible symmetry (6.3) for future work.

| Quantum Gate | Nontrivial transformation |
|:---:|:---:|
| $CZ_{i,j}$ | $X_i \to X_i Z_j,\ X_j \to X_j Z_i$ |
| $U_i^H$ | $X_i \to Z_i^\dagger,\ Z_i \to X_i$ |
| $S_{i,j}$ | $X_i \to X_j,\ X_j \to X_i,\ Z_i \to Z_j,\ Z_j \to Z_i$ |

Table 2: Nontrivial actions of the $\mathbb{Z}_N$ quantum gates.

# Acknowledgements

We would like to thank Yunqin Zheng for initial collaboration and for inspiring suggestions and discussions. We thank Shuheng Shao for useful comments after reading the draft. W.C. thanks Shinsei Ryu, Ramanjit Sohal, Sahand Seifnashri, Kantaro Ohmori, Shuheng Shao for useful discussions. W.C. is supported by the Global Science Graduate Course (GSGC) program of the University of Tokyo. W.C. also acknowledges support from JSPS KAKENHI grant numbers JP19H05810, JP22J21553 and JP22KJ1072. M.Y. is also supported in part by the JSPS Grant-in-Aid for Scientific Research (20H05860, 23K17689, 23K25865), and by JST, Japan (PRESTO Grant No. JPMJPR225A, Moonshot R&D Grant No. JPMJMS2061). He would also like to thank the Galileo Galilei Institute for Theoretical Physics for the hospitality and the INFN for partial support during the completion of this work. The authors of this paper were ordered alphabetically.

# A   Kramers-Wannier duality symmetry in lattice models

In this Appendix, we review KW transformation obtained by gauging a $\mathbb{Z}_N$ symmetry on a $1d$ spin chain. The main idea has been discussed e.g. in [5]. For reference, the definitions of $\mathbb{Z}_N$ unitary gates are

$$
\begin{aligned}
CZ_{i,j} &= \frac{1}{N} \sum_{\alpha,\beta=1}^{N} \omega^{-\alpha\beta} Z_i^\alpha Z_j^\beta, \\
U_i^H &= \frac{1}{\sqrt{N}} \sum_{\alpha,\beta=1}^{N} \omega^{-\alpha\beta} \left|\beta\right\rangle \left\langle\alpha\right|, \\
S_{i,j} &= \frac{1}{N} \sum_{\alpha,\beta=1}^{N} \omega^{\alpha\beta} (X_i^\alpha Z_i^\beta)(X_j^{-\alpha} Z_j^{-\beta}),
\end{aligned}
\tag{A.1}
$$

and their actions are summarized in table 2.

## A.1  Gauging $\mathbb{Z}_N$ symmetry on the whole spin chain

For simplicity, consider $\mathbb{Z}_N$ clock model at critical point on a chain with $L$ sites

$$H = -\sum_{i=1}^{L}(Z_{i-1}^{\dagger}Z_i + X_i) + (\text{h.c.}). \tag{A.2}$$

Since the $\mathbb{Z}_N$ symmetry $\eta = \prod_{i=1}^{L} X_i$ is onsite, and thus non-anomalous, we can gauge this symmetry by coupling the gauge variables $\tilde{X}_{i+1/2}, \tilde{Z}_{i+1/2}$ on the link. The gauged Hamiltonian is

$$H_{\text{gauged}} = -\sum_{i=1}^{L}(Z_{i-1}^{\dagger}\tilde{X}_{i-1/2}Z_i + X_i) + (\text{h.c.}). \tag{A.3}$$

which has a enlarged Hilbert space with dimension $N^{2L}$ and gauge symmetries generated by

$$G_i = \tilde{Z}_{i-1/2}X_i\tilde{Z}_{i+1/2}^{\dagger}, \quad [G_i, H_{\text{gauged}}] = 0, \quad \forall i. \tag{A.4}$$

We need to project to gauge-invariant sector by imposing Gauss law constraints

$$G_i = \tilde{Z}_{i-1/2}X_i\tilde{Z}_{i+1/2}^{\dagger} = 1, \quad \rightarrow \quad X_i = \tilde{Z}_{i-1/2}^{\dagger}\tilde{Z}_{i+1/2}, \quad \forall i. \tag{A.5}$$

The gauged Hamiltonian has a new 0-form $\mathbb{Z}_N$ symmetry which is easily seen by introducing a new set of Pauli operators

$$\hat{Z}_{i-1/2} = \tilde{Z}_{i-1/2}, \quad \hat{X}_{i-1/2} = Z_{i-1}^{\dagger}\tilde{X}_{i-1/2}Z_i. \tag{A.6}$$

The gauged Hamiltonian becomes

$$H_{\text{gauged}} = -\sum_{i=1}^{L}(\hat{X}_{i-1/2} + \hat{Z}_{i-1/2}^{\dagger}\hat{Z}_{i+1/2}) + (\text{h.c.}). \tag{A.7}$$

We recover the original Hamiltonian by renaming the variables

$$\hat{Z}_{i-1/2} \rightarrow Z_{i-1}, \quad \hat{X}_{i-1/2} \rightarrow X_{i-1}. \tag{A.8}$$

The whole process gives the famous KW transformation on the $\mathbb{Z}_N$-symmetric operator

$$Z_{i-1}^{\dagger}Z_i \rightarrow X_{i-1}, \quad X_i \rightarrow Z_{i-1}^{\dagger}Z_i. \tag{A.9}$$

The renaming is an isomorphism between two Hilbert spaces, both of which we denote by $\mathcal{H}$. Notice that the identification (A.9) involves a "half-translation" on the lattice, i.e., the fusion rule of KW duality operator will involve the one-site translation operator $\mathsf{T}$.

## A.2 Gauging $\mathbb{Z}_N$ symmetry on a half of the spin chain

Suppose we only gauge the $\mathbb{Z}_N$ symmetry on the half of the closed chain. Without loss of generality, we couple the gauge variables $\tilde{X}_i, \tilde{Z}_i$ on $J$ links from $(L, 1)$ to $(J - 1, J)$ with $J < L$. The half-gauged Hamiltonian is

$$H_{\text{half-gauged}} = -\sum_{i=1}^{J}(Z_{i-1}^\dagger \tilde{X}_{i-1/2} Z_i + X_i) - \sum_{i=J+1}^{L-1}(Z_{i-1}^\dagger Z_i + X_i) + (\text{h.c.}). \tag{A.10}$$

The dimension of the enlarged Hilbert space is $N^{L+J}$. However, there are only $J - 1$ generators of gauge symmetry

$$G_i := \tilde{Z}_{i-1/2}^\dagger X_i^\dagger \tilde{Z}_{i+1/2}, \quad i = 1, ..., J - 1, \tag{A.11}$$

that commute with the half-gauged Hamiltonian $H_{\text{half-gauged}}$. After imposing the $J - 1$ Gauss law constraints $G_i = 1, i = 1, ..., J - 1$ and projecting to the gauge invariant sector, the resulting Hilbert space has dimension $N^{J+1}$, with an extra degree of freedom compared with the original Hilbert space. We can then introduce a new set of Pauli variables $\hat{X}_i, \hat{Z}_i$ for the gauged half line

$$\begin{aligned}
\hat{Z}_{1/2} &:= \tilde{Z}_{1/2}, \quad \hat{X}_{1/2} := \tilde{X}_{1/2} Z_1, \\
\hat{Z}_{i-1/2} &:= \tilde{Z}_{i-1/2}, \quad \hat{X}_{i-1/2} := Z_{i-1}^\dagger \tilde{X}_{i-1/2} Z_{i-1}, \quad i = 2, ..., J - 2 \\
\hat{Z}_{J-1/2} &:= \tilde{Z}_{J-1/2}, \quad \hat{X}_{J-1/2} := Z_{J-1}^\dagger \tilde{X}_{1/2}.
\end{aligned} \tag{A.12}$$

The boundary terms are defined so that they have standard commutation relation with the operators at site $J$ and $L$. The gauged Hamiltonian then becomes

$$\begin{aligned}
H_{\text{half-gauged}} = &-(Z_L^\dagger \hat{X}_{1/2} + \hat{Z}_{1/2}^\dagger \hat{Z}_{3/2}) - \sum_{i=2}^{J-1}(\hat{X}_{i-1/2} + \hat{Z}_{i-1/2} \hat{Z}_{i+1/2}) \\
&-(\hat{X}_{J-1/2} Z_J + X_J) - \sum_{i=J+1}^{L-1}(Z_{i-1}^\dagger Z_i + X_i) + (\text{h.c.}).
\end{aligned} \tag{A.13}$$

After renaming $\hat{X}_{i-1/2} \to X_i, \hat{Z}_{i-1/2} \to Z_i$ for $i = 1, ..., J-2$ and $\hat{X}_{J-1/2} \to X_{(J-1,J)}, \hat{Z}_{J-1/2} \to Z_{(J-1,J)}$, which is the extra degree of freedom after imposing the Gauss law, the gauged Hamiltonian becomes

$$\begin{aligned}
H_{\text{half-gauged}} = &-(Z_L^\dagger X_1 + Z_1^\dagger Z_2) - \sum_{i=2}^{J-2}(X_i + Z_i^\dagger Z_{i+1}) \\
&-(X_{J-1} + Z_{J-1}^\dagger Z_{(J-1,J)} + X_{(J-1,J)} Z_J + X_J) \\
&-\sum_{i=J+1}^{L-1}(Z_{i-1}^\dagger Z_i + X_i) + (\text{h.c.}).
\end{aligned} \tag{A.14}$$

By conjugating the Hamiltonian with the unitary operator $CZ^\dagger_{(J-1,J),J}$, and reshuffling the terms, we get a simpler form

$$
\begin{aligned}
H_{\text{half-gauged}} = & - (Z^\dagger_L X_1) - \sum_{i=2}^{J-2} (Z^\dagger_{i-1} Z_i + X_i) \\
& - (Z^\dagger_{J-1} Z_{(J-1,J)} + X_{(J-1,J)} + Z^\dagger_{(J-1,J)} X_J) \\
& - \sum_{i=J+1}^{L-1} (Z^\dagger_{i-1} Z_i + X_i) + (\text{h.c.}).
\end{aligned}
$$

(A.15)

From the above expression, the gauged Hamiltonian is also given by the original Hamiltonian by inserting the KW defect D at the link $(L, 1)$ and the Hermitian conjugate $\mathsf{D}^\dagger$ at the link $(J - 1, J)$.

Now we can study the movement and fusion of the non-invertible topological defects. The defect Hamiltonian with insertion of a single KW defect D on the link $(J - 1, J)$ is

$$
H_{\mathsf{D}}^{(J-1,J)} = - \sum_{i=1}^{J-1} (Z^\dagger_{i-1} Z_i + X_i) - Z^\dagger_{J-1} X_J - \sum_{i=J+1}^{L-1} (Z^\dagger_{i-1} Z_i + X_i) + (\text{h.c.}),
$$

(A.16)

and we can use the unitary operator $U_{\mathsf{D}}^J = CZ^\dagger_{(J,J+1)} H^\dagger_J$, implementing the transformation

$$
\begin{aligned}
X_J \to Z_J, \quad Z_J \to X^\dagger_J Z_{J+1} \\
X_{J+1} \to Z^\dagger_J X_{J+1}, \quad Z_{J+1} \to Z_{J+1},
\end{aligned}
$$

(A.17)

to move this defect to the link $(J, J + 1)$.

To fuse the defects considering the insertion of two KW defects on the adjacent links $(J-1, J)$ and $(J, J + 1)$

$$
H_{\mathsf{D};\mathsf{D}}^{(J-1,J),(J,J+1)} = - \sum_{i=1}^{J-1} (Z^\dagger_{i-1} Z_i + X_i) - (Z^\dagger_{J-1} X_J + Z^\dagger_J X_{J+1}) - \sum_{i=J+1}^{L-1} (Z^\dagger_{i-1} Z_i + X_i) + (\text{h.c.}).
$$

(A.18)

The fusion operator $\lambda^J_{\mathsf{D}\otimes\mathsf{D}} = (U_{\mathsf{D}}^J)^\dagger$ will map this Hamiltonian to

$$
\begin{aligned}
H_{\mathsf{D}\otimes\mathsf{D}}^{(J-1,J+1)} = & \lambda^J_{\mathsf{D}\otimes\mathsf{D}} H_{\mathsf{D};\mathsf{D}}^{(J-1,J),(J,J+1)} (\lambda^J_{\mathsf{D}\otimes\mathsf{D}})^\dagger \\
= & - \sum_{i=1}^{J-1} (Z^\dagger_{i-1} Z_i + X_i) - (Z^\dagger_J Z^\dagger_{J-1} Z_{J+1} + X_{J+1}) - \sum_{i=J+1}^{L-1} (Z^\dagger_{i-1} Z_i + X_i) + (\text{h.c.}).
\end{aligned}
$$

(A.19)

In the fused defect Hamiltonian, the degrees of freedom on site $J$ is decoupled and is equivalent to an insertion of the $\mathsf{T}^{-1}$ defect [5], while $Z_J$ becomes a symmetry of the new Hamiltonian. The Hilbert space is a direct sum of the eigenspaces of $Z_J$. When $Z_J$ takes an eigenvalue $\omega^k$, the new Hamiltonian is equivalent to the original Hamiltonian with a defect $\eta^k$ inserted between $J - 1, J + 1$. Therefore the topological defect D follows the same fusion rule as its operator counterpart:

$$
\mathsf{D} \times \mathsf{D} = \left( \sum_{k=1}^N \eta^k \right) \mathsf{T}^{-1}.
$$

(A.20)

# B Topological defects of invertible symmetries in dipole Ising model

In this section, we review the topological defects of invertible symmetries $\eta_Q, \eta_D, \mathsf{C}$ in the dipole Ising model

$$H = -g^{-1} \sum_i Z_{i-1}(Z_i^\dagger)^2 Z_{i+1} - g \sum_i X_i + \text{(h.c.)}. \tag{B.1}$$

Much of the discussion can be generalized to other models with dipole symmetry. We study the defects through the defect Hamiltonian which is obtained by conjugating the original Hamiltonian with a twist operator, the symmetry transformation on a half of the infinite chain.

### $\eta_Q$ defect

The charge symmetry $\mathbb{Z}_N^Q$ is generated by

$$\eta_Q := \prod_i X_i, \quad \eta_Q^N = 1, \tag{B.2}$$

while the $\mathbb{Z}_N$ twist operator acting on $i \leq 0$ is

$$\eta_{Q,0} = \prod_{i \leq 0} X_i. \tag{B.3}$$

The defect Hamiltonian with a $\eta_Q$ defect at the link (0,1) is

$$\begin{aligned} H_Q^{(0,1)} &= \eta_{Q,0} H \eta_{Q,0}^\dagger \\ &= -g^{-1} \sum_{i \neq 0,1} Z_{i-1}(Z_i^\dagger)^2 Z_{i+1} - g \sum_i X_i \\ &\quad - g^{-1} \omega Z_{-1}(Z_0^\dagger)^2 Z_1 - g^{-1} \omega^{-1} Z_0 (Z_1^\dagger)^2 Z_2 + \text{(h.c.)}, \end{aligned} \tag{B.4}$$

where we label the defect Hamiltonian with the type and location of defect. We also show the movement operators, which are local unitary operators and move the defect by one site. For example, the movement operator for the $\eta_Q$ defect is

$$U_Q^1 := X_1 : \quad H_Q^{(0,1)} \to H_Q^{(1,2)} = U_\eta^1 H_Q^{(0,1)} (U_Q^1)^\dagger. \tag{B.5}$$

Since the movement operators are local unitary operators, the movement does not cost energy and the defect is topological. One can also put several defects and fuse them. The fusion of topological defects shares the same rule of the fusion of corresponding operators.

### $\eta_D$ defect

The dipole symmetry $\mathbb{Z}_N^D$ is generated by

$$\eta_D = \prod_i X_i^i, \quad \eta_D^N = 1. \tag{B.6}$$

The twist operator acting on sites $i < i_0$ is

$$\eta_{D,i_0-1} = \prod_{i<i_0} (X_i)^i, \tag{B.7}$$

with the movement operator $(X_{i_0})^{i_0}$. The defect Hamiltonian with a $\eta_D$ defect inserted at the link $(i_0 - 1, i_0)$ is

$$\begin{aligned}
H_D^{(i_0-1,i_0)} = &- g^{-1} \sum_{i \neq i_0-1, i_0} Z_{i-1}(Z_i^\dagger)^2 Z_{i+1} - g \sum_i X_i \\
&- g^{-1}\omega^{-i_0} Z_{i_0-2}(Z_{i_0-1}^\dagger)^2 Z_{i_0} - g^{-1}\omega^{i_0-1} Z_{i_0-1}(Z_{i_0}^\dagger)^2 Z_{i_0+1} + \text{(h.c.)}.
\end{aligned} \tag{B.8}$$

**Charge conjugation defect**

The charge conjugation symmetry $\mathbb{Z}_2^C$ is generated by

$$\mathsf{C} = \prod_i \mathsf{C}_i, \quad \mathsf{C}_i: \quad Z_i \to Z_i^\dagger, \quad X_i \to X_i^\dagger, \tag{B.9}$$

The twist operator is the truncation of $\mathsf{C}$ on half of the chain and the movement operator is $\mathsf{C}_i$. The defect Hamiltonian with a charge conjugation defect inserted at link $(0, 1)$ is

$$\begin{aligned}
H_{\mathsf{C}}^{(0,1)} = &- g^{-1} \sum_{i \neq 0,1} Z_{i-1}(Z_i^\dagger)^2 Z_{i+1} - g \sum_i X_i \\
&- g^{-1} Z_{-1}(Z_0^\dagger)^2 Z_1^\dagger - g^{-1} Z_0^\dagger (Z_1^\dagger)^2 Z_2 + \text{(h.c.)}.
\end{aligned} \tag{B.10}$$

# C  Computation in bilinear phase map representation

In this appendix, we compute all the operator relations and fusions rigorously in the BMP representation. For reference, we list the action of Pauli operators on bra and ket states

$$\begin{aligned}
Z_i |s_i\rangle = \sum_{\alpha=1}^N \omega^\alpha |\alpha\rangle \langle \alpha | s_i\rangle = \omega^{s_i} |s_i\rangle, \quad \langle s_i| Z_i = \sum_{\alpha=1}^N \omega^\alpha \langle s_i|\alpha\rangle \langle \alpha| = \omega^{s_i} \langle s_i|, \\
X_i |s_i\rangle = \sum_{\alpha=1}^N |\alpha+1\rangle \langle \alpha | s_i\rangle = |s_i+1\rangle, \quad \langle s_i| X_i = \sum_{\alpha=1}^N \langle s_i|\alpha+1\rangle \langle \alpha| = \langle s_i-1|.
\end{aligned} \tag{C.1}$$

The bra and ket states are eigenstates of the $Z_i$, while $X_i$ shifts the ket state by $+1$ and the bra state by $-1$. Starting from a state $|\{s_i\}\rangle = \otimes_{i=1}^L |s_i\rangle_i$, we define the state after a lattice translation as $|\{s_{i-1}\}\rangle := \mathsf{T} |\{s_i\}\rangle := \otimes_{i=1}^L |s_{i-1}\rangle_i$, where all values of qudit on each site has been move to one site right. The lattice translation operator is therefore defined as

$$\mathsf{T} = \sum_{\{s_i\}} |\{s_{i-1}\}\rangle \langle \{s_i\}| \tag{C.2}$$

and its action on Pauli operators is

$$\mathsf{T} Z_i \mathsf{T}^{-1} = Z_{i+1}, \quad \mathsf{T} X_i \mathsf{T}^{-1} = X_{i+1}. \tag{C.3}$$

## KW transformation

The KW transformation is given by

$$\mathsf{D} = \sum_{\{s_i\},\{s_i'\}} \omega^{\sum_{i=1}^{L}(s_{i-1}-s_i)s_{i-1}'} |\{s_i'\}\rangle \langle\{s_i\}| = \sum_{\{s_i\},\{s_i'\}} \omega^{\sum_{i=1}^{L}(s_i'-s_{i-1}')s_i} |\{s_i'\}\rangle \langle\{s_i\}| . \qquad (C.4)$$

The transformation of Pauli operators is

$$\begin{aligned}
\mathsf{D}\left(Z_{j-1}^{\dagger}Z_j\right) &= \sum_{\{s_i\},\{s_i'\}} \omega^{\sum_{i=1}^{L}(s_{i-1}-s_i)s_{i-1}'} |\{s_i'\}\rangle \langle\{s_i\}| \left(Z_{j-1}^{\dagger}Z_j\right) \\
&= \sum_{\{s_i\},\{s_i'\}} \omega^{\sum_{i=1}^{L}(s_{i-1}-s_i)s_{i-1}'} |\{s_i'\}\rangle \langle\{s_i\}| \,\omega^{-(s_{j-1}-s_j)} \\
&= \sum_{\{s_i\},\{s_i'\}} \omega^{\sum_{i\neq j-1}(s_{i-1}-s_i)s_{i-1}'+(s_{j-1}-s_j)(s_{j-1}'-1)} |\{s_i'\}\rangle \langle\{s_i\}| \qquad (C.5) \\
&= \sum_{\{s_i\},\{s_i'\}} \omega^{\sum_i (s_{i-1}-s_i)s_{i-1}'} |\{s_{i\neq j-1}'; s_{j-1}'+1\}\rangle \langle\{s_i\}| \\
&= X_{j-1}\mathsf{D},
\end{aligned}$$

where we rename $s_{j-1}' - 1$ by $\hat{s}_{j-1}'$ after the third equality and drop the hat in the fourth equality. The notation $|\{s_{i\neq j-1}'; s_{j-1}'+1\}\rangle$ differs from $|\{s_i'\}\rangle$ only by the shift of qudit on the $j-1$ site. Similarly,

$$\begin{aligned}
\mathsf{D}X_j &= \sum_{\{s_i\},\{s_i'\}} \omega^{\sum_{i=1}^{L}(s_i'-s_{i-1}')s_i} |\{s_i'\}\rangle \langle\{s_i\}| \, X_j \\
&= \sum_{\{s_i\},\{s_i'\}} \omega^{\sum_{i=1}^{L}(s_i'-s_{i-1}')s_i} |\{s_i'\}\rangle \langle\{s_{i\neq j}; s_j-1\}| \\
&= \sum_{\{s_i\},\{s_i'\}} \omega^{\sum_{i\neq j}(s_i'-s_{i-1}')s_i+(s_j-s_{j-1})(s_j'+1)} |\{s_i'\}\rangle \langle\{s_i\}| \qquad (C.6) \\
&= Z_{j-1}^{\dagger}Z_j\mathsf{D}.
\end{aligned}$$

The fusion of translation operator with the KW operator is

$$\begin{aligned}
\mathsf{T}\mathsf{D} &= \sum_{\{s_i\},\{s_i'\}} \omega^{\sum_{i=1}^{L}(s_i'-s_{i-1}')s_i} T |\{s_i'\}\rangle \langle\{s_i\}| \\
&= \sum_{\{s_i\},\{s_i'\}} \omega^{\sum_{i=1}^{L}(s_i'-s_{i-1}')s_i} |\{s_{i-1}'\}\rangle \langle\{s_i\}| \\
&= \sum_{\{s_i\},\{\hat{s}_i\}} \omega^{\sum_{i=1}^{L}(\hat{s}_{i+1}-\hat{s}_i)s_i} |\{\hat{s}_i\}\rangle \langle\{s_i\}| \qquad (C.7) \\
&= \sum_{\{s_i\},\{\hat{s}_i\}} \omega^{\sum_{i=1}^{L}-(\hat{s}_i-\hat{s}_{i+1})s_i} |\{\hat{s}_i\}\rangle \langle\{s_i\}| ,
\end{aligned}$$

and the conjugate of D is

$$D^\dagger = \sum_{\{s_i\},\{s_i'\}} \omega^{\sum_{i=1}^{L} -(s_{i-1}-s_i)s_{i-1}'} |\{s_i\}\rangle \langle\{s_i'\}| . \tag{C.8}$$

Therefore, by renaming the dummy variables we show that $TD = D^\dagger$. Similarly,

$$
\begin{aligned}
DT &= \sum_{\{s_i\},\{s_i'\}} \omega^{\sum_{i=1}^{L}(s_i'-s_{i-1}')s_i} |\{s_i'\}\rangle \langle\{s_i\}| T \\
&= \sum_{\{s_i\},\{s_i'\}} \omega^{\sum_{i=1}^{L}(s_i'-s_{i-1}')s_i} |\{s_i'\}\rangle \langle\{s_{i+1}\}| \\
&= \sum_{\{s_i'\},\{\hat{s}_i\}} \omega^{\sum_{i=1}^{L}(s_i'-s_{i-1}')\hat{s}_{i-1}} |\{s_i'\}\rangle \langle\{\hat{s}_i\}| \\
&= \sum_{\{s_i\},\{\hat{s}_i\}} \omega^{\sum_{i=1}^{L} -(s_{i-1}'-s_i')\hat{s}_{i-1}} |\{s_i'\}\rangle \langle\{\hat{s}_i\}| = D^\dagger .
\end{aligned}
\tag{C.9}
$$

Therefore we confirmed that the translation operator commutes with the KW operator and their fusion is the Hermitian conjugate of the KW operator:

$$D^\dagger = TD = DT. \tag{C.10}$$

The non-invertible fusion rule is given by

$$
\begin{aligned}
D \times D &= \sum_{\{s_i\},\{s_i'\},\{m_i\},\{m_i'\}} \omega^{\sum_{i=1}^{L}(s_i'-s_{i-1}')s_i} \omega^{\sum_{i=1}^{L}(m_{i-1}-m_i)m_{i-1}'} |\{s_i'\}\rangle \langle\{s_i\}|\{m_i'\}\rangle \langle\{m_i\}| \\
&= \sum_{\{s_i\},\{s_i'\},\{m_i\}} \omega^{\sum_{i=1}^{L}(s_i'-s_{i-1}'+m_i-m_{i+1})s_i} |\{s_i'\}\rangle \langle\{m_i\}| \\
&= \sum_{\{s_i'\},\{m_i\}} \prod_{i=1}^{L} \delta_{s_i'-s_{i-1}'+m_i-m_{i+1},0} |\{s_i'\}\rangle \langle\{m_i\}| .
\end{aligned}
\tag{C.11}
$$

The solution for the constraints $s_i' - s_{i-1}' + m_i - m_{i+1} = 0, i = 1, .., L$ is

$$s_i' = m_{i+1} + k, \quad k = 1, ..., N, i = 1, ..., L. \tag{C.12}$$

Therefore, the fusion rule is

$$
\begin{aligned}
D \times D &= \sum_{\{m_i\}} \sum_{k=1}^{N} |\{m_{i+1} + k\}\rangle \langle\{m_i\}| \\
&= \left(\sum_{k=1}^{N} \eta^k\right) \sum_{\{m_i\}} |\{m_{i+1}\}\rangle \langle\{m_i\}| \\
&= \left(\sum_{k=1}^{N} \eta^k\right) T^{-1} .
\end{aligned}
\tag{C.13}
$$

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
