# Peer review of "Generating Lattice Non-invertible Symmetries"

_SciPost Physics_

## Round 2 · Referee Report · Anonymous (Referee 5) · 2024-8-4

Strengths

Uncovers an interesting interplay between a dipole symmetry and a non-invertible symmetry in concrete lattice models.

Weaknesses

None

Report

The paper discusses aspects of non-invertible symmetries of lattice models with tensor product Hilbert space. One of the main results is the discovery of a dipole Kramers-Wannier (KW) non-invertible symmetry. The corresponding symmetry operator can be obtained by conjugating a unitary operator with a seed non-invertible symmetry, and it implements the gauging of a dipole symmetry. By using the lattice half-space gauging of dipole symmetry, the defect Hamiltonian twisted by the dipole KW symmetry is also obtained.

The authors further show that certain dipole KW symmetries are anomalous (i.e., incompatible with a symmetric trivially gapped phase) by explicitly scanning through the known dipole symmetry protected topological phases. Their finding is compatible with the phase diagram of the dipole Ising model, which is known from numerical simulations. Finally, they worked out the generalized Kennedy-Tasaki transformation for the new dipole KW symmetry.

The paper is clearly written, and contains new interesting results on lattice models with generalized symmetries. It provides a bridge between two topics in the field which are often studied separately, non-invertible symmetries and modulated symmetries such as the dipole symmetry. Therefore, the referee recommends that the paper is published in SciPost.

Requested changes

Please find below minor comments and questions:

- In the sentence below Eq. (2.13), it is said that "the subalgebra $\{ \eta, \mathsf{D}\}$ differs from the fusion algebra of $\mathsf{TY}(\mathbb{Z}_N)$." Strictly speaking, $\eta$ and $\mathsf{D}$ do not form a subalgebra since the algebra doesn't close between them without involving the translation. Would it be more precise to say that "the lattice symmetry algebra involving $\eta$ and $\mathsf{D}$ differs from the fusion algebra of $\mathsf{TY}(\mathbb{Z}_N)$"?

- It is found that the algebra of the dipole KW symmetry operator involves the charge conjugation symmetry. Just as a curious comment, this seems to resemble the algebra of non-invertible duality symmetries found in 3+1d continuum field theories. See, e.g., Eq. (1.4) of https://arxiv.org/pdf/2204.09025.

- On page 3, it is mentioned that the non-invertible chiral symmetry in 3+1d "is constructed by stacking a (2+1)d fractional qunatum Hall state to a *gauge-dependent conserved current." To be more precise, the non-invertible operator there is obtained by stacking the FQHE state to a gauge-invariant current which is not conserved.

Recommendation

Publish (easily meets expectations and criteria for this Journal; among top 50%)

  • validity: -
  • significance: -
  • originality: -
  • clarity: -
  • formatting: -
  • grammar: -

Author:  Weiguang Cao  on 2024-08-10  [id 4688]

(in reply to Report 1 on 2024-08-04)
Category:
correction

Thanks for the careful reading and kind suggestions. We will response to each requested change in the following

  • In the sentence below Eq. (2.13), it is said that "the subalgebra {η,D} differs from the fusion algebra of TY(ZN)." Strictly speaking, η and D do not form a subalgebra since the algebra doesn't close between them without involving the translation. Would it be more precise to say that "the lattice symmetry algebra involving η and D differs from the fusion algebra of TY(ZN)"?

Response: We rephrased the sentence as the referee suggested.

  • It is found that the algebra of the dipole KW symmetry operator involves the charge conjugation symmetry. Just as a curious comment, this seems to resemble the algebra of non-invertible duality symmetries found in 3+1d continuum field theories. See, e.g., Eq. (1.4) of https://arxiv.org/pdf/2204.09025.

Response: We added the resemblance of the algebra of non-invertible duality symmetries in 3+1d continuum field theories in footnote 7 on page 4.

  • On page 3, it is mentioned that the non-invertible chiral symmetry in 3+1d "is constructed by stacking a (2+1)d fractional qunatum Hall state to a *gauge-dependent conserved current." To be more precise, the non-invertible operator there is obtained by stacking the FQHE state to a gauge-invariant current which is not conserved.

Response: We corrected the statement as the referee suggested.

---

## Round 2 · Referee Report · Anonymous (Referee 4) · 2024-8-5

Strengths

The paper is almost fully self-contained, and comprehensive for both quantum matter and hep-th community.

Report

This paper discusses a method of generating lattice non-invertible symmetries by composing seed non-invertible transformations, or conjugating unitary transformation by seed non-invertible transformations. In particular, the authors construct the non-invertible dipole Kramers-Wannier (KW) symmetry, from conjugating Hadamard transformation by a seed KW transformation. The fusion algebra of this non-invertible symmetry, along with its symmetry defect and anoalies are discussed, with detailed lattice analysis. The generalized Kennedy-Tasaki transformation between dipole SSB phases and dipole SPT phases is further constructed with the help of dipole KW symmetry.

This paper focuses on the intersection of two novel symmetries (dipole symmetries and non-invertible symmetries), and discusses interesting examples. I happily recommend it for publication.

Requested changes

1- On page 4, it is mentioned that gauging the dipole symmetry and gauging the $\mathbb{Z}_N^Q\times \mathbb{Z}_N^D$ symmetry are just different terms for the same procedure. I'm wondering if this fact can be understood in the algebra between $D$ (KW trasnsformation) and $\hat{D}$ (dipole KW transformation), e.g. is $D\hat{D}=\hat{D}$?

2- The XZ model introduced in Eq. (2.20) has more symmetries than just charge $\mathbb{Z}_N$ mentioned above the equation. For example, there is a Hadamard symmetry and another $\mathbb{Z}_N$ symmetry. The former symmetry becomes the dipole KW after gauging the charge symmetry. The latter symmetry is mentioned later on page 21. I suggest a more detailed discussion on the symmetries in this model, and how the symmetries become the dipole-related symmetries after gauging the charge $\mathbb{Z}_N$ symmetry. This might help to understand the sandwich construction in general.

3- On page 12, the gauging of dipole symmetry is performed on a dipole-Ising model. However, I think it is not clearly showed in the paper how Eq. (3.3) and Eq. (3.4) are obtained. Furthermore, after a change of variables in Eq. (3.7), some discussions are needed before the authors ignore the Gauss law constraints and claim that the gauged model is again a dipole-Ising model on a tensor-product Hilbert space.

Recommendation

Publish (easily meets expectations and criteria for this Journal; among top 50%)

  • validity: high
  • significance: high
  • originality: top
  • clarity: high
  • formatting: perfect
  • grammar: perfect

Author:  Weiguang Cao  on 2024-08-10  [id 4687]

(in reply to Report 2 on 2024-08-05)
Category:
answer to question

Thanks for the careful reading and kind suggestions. We will response to each requested change in the following

1- On page 4, it is mentioned that gauging the dipole symmetry and gauging the $Z^Q_N×Z^D_N$ symmetry are just different terms for the same procedure. I'm wondering if this fact can be understood in the algebra between $\mathsf{D}$ (KW trasnsformation) and $\hat{\mathsf{D}}$ (dipole KW transformation), e.g. is $\mathsf{D}\hat{\mathsf{D}}=\hat{\mathsf{D}}$?

Response: in our paper, we focused on the duality transformation by gauging the $Z^Q_N×Z^D_N$ symmetry. We sometimes abuses the terminology 'gauging the dipole symmetry’ which also happens in the field theory where people gauge both symmetry but called it ‘dipole gauge symmetry’. $\mathsf{D}\hat{\mathsf{D}}=\hat{\mathsf{D}}$ is not quite correct because starting from the XZ model, the duality transformation $\mathsf{D}\hat{\mathsf{D}}$ leads to the dipole ising model while the duality $\hat{\mathsf{D}}$ leads to the XZ model.

2- The XZ model introduced in Eq. (2.20) has more symmetries than just charge $Z_N$ mentioned above the equation. For example, there is a Hadamard symmetry and another $Z_N$ symmetry. The former symmetry becomes the dipole KW after gauging the charge symmetry. The latter symmetry is mentioned later on page 21. I suggest a more detailed discussion on the symmetries in this model, and how the symmetries become the dipole-related symmetries after gauging the charge $Z_N$ symmetry. This might help to understand the sandwich construction in general.

Response: The XZ model only has the Hadamard symmetry when $g=1$. For the other $Z_N$ symmetry generated by $\prod_{i=1}^L Z_i$, it indeed becomes the dipole symmetry in the dipole Ising model. We added a comment around Eq. (2.29) in page 10 to show how this happens under the KW duality transformation.

3- On page 12, the gauging of dipole symmetry is performed on a dipole-Ising model. However, I think it is not clearly showed in the paper how Eq. (3.3) and Eq. (3.4) are obtained. Furthermore, after a change of variables in Eq. (3.7), some discussions are needed before the authors ignore the Gauss law constraints and claim that the gauged model is again a dipole-Ising model on a tensor-product Hilbert space.

Response: we add the details about promoting the global symmetry to gauge symmetry below Eq.(3.2). Now the Gauss law condition can be seen directly from the gauge transformation (new Eq.(3.7) ). About the second point, we didn’t ignore the Gauss law constraints but impose the constraints to reduce the enlarged Hilbert space of the gauge theory to a new Hilbert space with dimension $2^L$. To highlight this point, we added a sentence “Afte imposing the Gauss law constraints, the enlarged Hilbert space $\tilde{\mathcal{H}}$ is projected down to a new Hilbert space with dimension $2^{L}$, which is isomorphic to the original Hilbert space $\mathcal{H}$.”

---

## Editorial Decision

resubmitted